# Uniform Wrappers: Bridging Concave to Quadratizable Functions in Online Optimization

## Abstract

This paper presents novel contributions to the field of online optimization, particularly focusing on the adaptation of algorithms from concave optimization to more challenging classes of functions. Key contributions include the introduction of uniform wrappers, establishing a vital link between upper-quadratizable functions and algorithmic conversions. Through this framework, the paper demonstrates superior regret guarantees for various classes of up-concave functions under zeroth-order feedback. Furthermore, the paper extends zeroth-order online algorithms to bandit feedback counterparts and offline counterparts, achieving a notable improvement in regret/sample complexity compared to existing approaches.

## 1 Introduction

The optimization of continuous DR-submodular functions has become increasingly prominent in recent years. This form of optimization represents an important subset of non-convex optimization problems at the forefront of machine learning and statistics. These challenges have numerous real-world applications like revenue maximization, mean-field inference, and recommendation systems, among others (Bian et al., 2019; Hassani et al., 2017; Mitra et al., 2021; Djolonga & Krause, 2014; Ito & Fujimaki, 2016; Gu et al., 2023; Li et al., 2023).

A natural staring point to for DR-submodular maximization is to start from a convex optimization algorithm and adapt it to the setting of DR-submodular functions. Online Convex Optimization (OCO) is extensively utilized across various fields due to its numerous practical applications and robust theoretical underpinnings. The tools from the area of online convex optimization have been applied to many online non-concave optimization algorithms, e.g., to converge to stationary points in online non-concave optimization (Yang et al., 2018), or algorithms with approximation guarantees for DR-submodular optimization (Chen et al., 2018; Niazadeh et al., 2020; Zhang et al., 2022; Pedramfar et al., 2023).

In this paper, we focus on a large class of functions, namely the class of quadratizable functions, first introduced in (Pedramfar & Aggarwal, 2024a). Quadratizable functions includes special subclasses of non-convex/non-concave functions where the offline constrained optimization problem is NP-hard to solve but we can find an $\alpha$-approximation of the optimal value in polynomial time. Indeed, it is shown that this class of online upper quadratizable optimization includes up-concave optimization (a generalization of DR-submodular and concave optimization) in the following cases: (i) monotone $\gamma$-weakly $\mu$-strongly DR-submodular functions with curvature $c$ over general convex sets, (ii) monotone $\gamma$-weakly DR-submodular functions over convex sets containing the origin, and (iii) non-monotone DR-submodular optimization over general convex sets.

Even though the tools from OCO have proven effective in more challenging classes, much of past work along these lines involve taking inspiration from OCO and manually designing new algorithms and analyzing them specific to each problem setting. This raises the following question

> *When and how can we adapt algorithms from the (simpler) setup of online convex optimization into algorithms for online optimization over more general classes of functions?*

In this paper, we try to provide partial solutions to this question for adapting OCO algorithms to algorithms for online quadratizable optimization. The notion of quadratizability is built upon a generalization of the defining condition $f(\mathbf{x}) - f(\mathbf{y}) \geq \langle \nabla f(\mathbf{y}), \mathbf{x} - \mathbf{y} \rangle$ of convex functions. This similarity with convex functions is a starting point which allows us to define a class of meta-algorithm called "uniform wrappers". Uniform wrappers provide a straightforward way to convert OCO algorithms into algorithms that can handle quadratizable functions. We also develop a guideline to convert the existing proofs for regret bounds of the base algorithms in the convex setting into regret bounds of the new algorithms over the quadratizable functions.

We note that, for a specific class of algorithms, this question was partly addressed in (Pedramfar & Aggarwal, 2024a). Specifically, as we will discuss in Appendix B, their result can be formulated as a special case of ours, where they assume that the starting algorithm is a first order online algorithm with semi-bandit feedback that obtains sub-linear regret against fully adaptive adversaries. This condition is too restrictive to allow for adapting many of the ideas in OCO literature. In this paper we take a step further and can handle broader classes of algorithms, including the more challenging setting of zeroth order feedback.

As an application of our framework, we propose a variant of a bandit convex optimization algorithm that was introduced in (Saha & Tewari, 2011) as the base algorithm, namely Zeroth Order Follow the Regularized Leader (ZO-FTRL) and demonstrate how it can be converted using uniform wrappers (denoted by $\mathcal{W}$) to obtain 3 algorithms for function classes (i)-(iii) mentioned above. See Tables 1 and 2 for details. Note that ZO-FTRL and $\mathcal{W}$(ZO-FTRL) are zeroth order, but they are not bandit feedback algorithms. We also extend the results to those with bandit feedback, as well as derive sample complexity guarantees for the offline algorithm.

The main contributions in this work include:

1. We develop a general framework for converting algorithms and their regret guarantees from online convex concave optimization to online quadratizable optimization. Conversion of the algorithm could be applied to any online optimization algorithm, and the conversion of the proof is described using a general guideline.

2. Our framework obtains or matches the state of the art algorithm in all online optimization settings considered. (See Table 1) Note that our framework also recovers all known results for non-stationary DR-submodular maximization. (See Remark 6 and Table 3 in (Pedramfar & Aggarwal, 2024a))

3. Except for deterministic first order feedback and the special case of $\gamma$-weakly non-monotone functions with $\gamma < 1$, our framework obtains or matches the state of the art algorithm in all online optimization settings considered. (See Table 2)

4. We obtain superior regret guarantees for several classes of weakly DR-submodular functions under zeroth order feedback, specifically (i) monotone $\gamma$-weakly $\mu$-strongly DR-submodular functions with curvature $c$ over general convex sets, (ii) monotone $\gamma$-weakly DR-submodular functions over convex sets containing the origin, and (iii) non-monotone DR-submodular optimization over general convex sets. (See Table 1 and Theorem 6)

5. Those results can be extended to the bandit setting yielding improved results for bandit feedback. (See Table 1 and Theorem 7)

6. The results for zeroth order online algorithms can be specialized to offline algorithms, resulting in three new algorithms with a sample complexity of $1/\epsilon^3$ in different settings, which is significantly better than the state of art $1/\epsilon^4$. (See Table 2 and Theorem 8)

 *To simplify the notation and statements, we define regret for maximization problems and focus on concave maximization and DR-submodular maximization.*

## 2 BACKGROUND AND NOTATION

For a set $\mathcal{D} \subseteq \mathbb{R}^d$, we define its *affine hull* $\mathrm{aff}(\mathcal{D})$ to be the set of $\alpha\mathbf{x} + (1-\alpha)\mathbf{y}$ for all $\mathbf{x}, \mathbf{y}$ in $\mathcal{K}$ and $\alpha \in \mathbb{R}$. The *relative interior* of $\mathcal{D}$ is defined as $\mathrm{relint}(\mathcal{D}) := \{\mathbf{x} \in \mathcal{D} \mid \exists r > 0, \mathbb{B}_r(\mathbf{x}) \cap \mathrm{aff}(\mathcal{D}) \subseteq \mathcal{D}\}$. All convex functions are continuous on any point in the relative interior of their domains. In this work, we will only focus on continuous functions. If $\mathbf{x} \in \mathrm{relint}(\mathcal{K})$ and $f$ is convex and is

Table 1: Online up-concave maximization

| F | Set | | Feedback | | Reference | Appx. | # of queries | $\log_T(\alpha\text{-regret})$ |
|---|---|---|---|---|---|---|---|---|
| Monotone | $0 \in \mathcal{K}$ | $\nabla F$ | Full Information | stoch. | (Zhang et al., 2022) (*) | $1-e^{-\gamma}$ | 1 | 1/2 |
| | | | | | (Pedramfar et al., 2024a) | $1-e^{-1}$ | $T^\theta(\theta \in [0,1/2])$ | $2/3-\theta/3$ |
| | | | | | (Pedramfar & Aggarwal, 2024a) (*) | $1-e^{-\gamma}$ | 1 | 1/2 |
| | | | Semi-bandit | stoch. | (Pedramfar et al., 2024a) | $1-e^{-1}$ | - | 3/4 |
| | | | | | (Pedramfar & Aggarwal, 2024a) (*) | $1-e^{-\gamma}$ | - | 2/3 |
| | | $F$ | Full Information | det. | (Pedramfar & Aggarwal, 2024a) (*) | $1-e^{-1}$ | 2 | 1/2 |
| | | | | stoch. † | Theorem 6 | $1-e^{-1}$ | 1 | 2/3 |
| | | | | stoch. | (Pedramfar et al., 2024a) | $1-e^{-1}$ | $T^\theta(\theta \in [0,1/4])$ | $4/5-\theta/5$ |
| | | | | | (Pedramfar & Aggarwal, 2024a) (*) | $1-e^{-\gamma}$ | 1 | 3/4 |
| | | | Bandit | det. | (Wan et al., 2023) (*) | $1-e^{-1}$ | - | 3/4 |
| | | | | | (Zhang et al., 2024) (*) | $1-e^{-\gamma}$ | - | 4/5 |
| | | | | stoch.† | Theorem 7 | $1-e^{-1}$ | - | 3/4 |
| | | | | stoch. | (Pedramfar et al., 2024a) | $1-e^{-1}$ | - | 5/6 |
| | | | | | (Pedramfar & Aggarwal, 2024a) (*) | $1-e^{-\gamma}$ | - | 4/5 |
| | general | $\nabla F$ | Full Information | stoch. | (Pedramfar et al., 2024a) | 1/2 | $T^\theta(\theta \in [0,1/2])$ | $2/3-\theta/3$ |
| | | | Semi-bandit | stoch. | (Chen et al., 2018) (*) | $\gamma^2/(1+\gamma^2)$ | - | 1/2 |
| | | | | | (Pedramfar et al., 2024a) | 1/2 | - | 3/4 |
| | | | | | (Pedramfar & Aggarwal, 2024a) (*) | $\gamma^2/(1+c\gamma^2)$ | - | 1/2 |
| | | $F$ | Full Information | det. | (Pedramfar & Aggarwal, 2024a) (*) | $\gamma^2/(1+c\gamma^2)$ | 2 | 1/2 |
| | | | | stoch. | (Pedramfar et al., 2024a) | 1/2 | $T^\theta(\theta \in [0,1/4])$ | $4/5-\theta/5$ |
| | | | | | Theorem 6 | $\gamma^2/(1+c\gamma^2)$ | 1 | 2/3 |
| | | | Bandit | stoch. | (Pedramfar et al., 2024a) | 1/2 | - | 5/6 |
| | | | | | (Pedramfar & Aggarwal, 2024a) (*) | $\gamma^2/(1+c\gamma^2)$ | - | 3/4 |
| | | | | | Theorem 7 | $\gamma^2/(1+c\gamma^2)$ | - | 3/4 |
| Non-Monotone | general | $\nabla F$ | Full Information | stoch. | (Pedramfar et al., 2024a) | $(1-h)/4$ | $T^\theta(\theta \in [0,1/2])$ | $2/3-\theta/3$ |
| | | | | | (Zhang et al., 2024) (*) | $(1-h)/4$ | 1 | 1/2 |
| | | | | | (Pedramfar & Aggarwal, 2024a) (*) | $(1-h)/4$ | 1 | 1/2 |
| | | | Semi-bandit | stoch. | (Pedramfar et al., 2024a) | $(1-h)/4$ | - | 3/4 |
| | | | | | (Pedramfar & Aggarwal, 2024a) (*) | $(1-h)/4$ | - | 2/3 |
| | | $F$ | Full Information | det. | (Pedramfar & Aggarwal, 2024a) (*) | $(1-h)/4$ | 2 | 1/2 |
| | | | | stoch. † | Theorem 6 | $(1-h)/4$ | 1 | 2/3 |
| | | | | stoch. | (Pedramfar et al., 2024a) | $(1-h)/4$ | $T^\theta(\theta \in [0,1/4])$ | $4/5-\theta/5$ |
| | | | | | (Pedramfar & Aggarwal, 2024a) (*) | $(1-h)/4$ | 1 | 3/4 |
| | | | Bandit | det. | (Zhang et al., 2024) (*) | $(1-h)/4$ | - | 4/5 |
| | | | | stoch. † | Theorem 7 | $(1-h)/4$ | - | 3/4 |
| | | | | stoch. | (Pedramfar et al., 2024a) | $(1-h)/4$ | - | 5/6 |
| | | | | | (Pedramfar & Aggarwal, 2024a) (*) | $(1-h)/4$ | - | 4/5 |

This table compares different static regret results for the online up-concave maximization. The logarithmic terms in regret are ignored. Here $h := \min_{\mathbf{z} \in \mathcal{K}} \|\mathbf{z}\|_\infty$. **Rows marked with (*) are results in the literature that are special cases of the results stated here and therefore fit within the framework described in this paper.** The rows describing results with stochastic feedback that are marked with † assume that the random query oracle is contained with a cone, as detailed in Theorem 6.

differentiable at $\mathbf{x}$, then we have $f(\mathbf{y}) - f(\mathbf{x}) \geq \langle \nabla f(\mathbf{x}), \mathbf{y} - \mathbf{x} \rangle$, for all $\mathbf{y} \in \mathcal{K}$. More generally, given $\mu \geq 0$, we say a vector $\mathbf{o} \in \mathbb{R}^d$ is a $\mu$-*subgradient* of $f$ at $\mathbf{x}$ if $f(\mathbf{y}) - f(\mathbf{x}) \geq \langle \mathbf{o}, \mathbf{y} - \mathbf{x} \rangle + \frac{\mu}{2} \|\mathbf{y} - \mathbf{x}\|^2$. for all $\mathbf{y} \in \mathcal{K}$. Given a convex set $\mathcal{K}$, a function $f : \mathcal{K} \to \mathbb{R}$ is $\mu$-strongly convex if and only if it has a $\mu$-subgradient at all points $\mathbf{x} \in \mathcal{K}$. A function $F : \mathcal{D} \to \mathbb{R}^+$ is $G$-*Lipschitz continuous* if for all $\mathbf{x}, \mathbf{y} \in \mathcal{D}$, $\|F(\mathbf{x}) - F(\mathbf{y})\| \leq G\|\mathbf{x} - \mathbf{y}\|$. A differentiable function $F : \mathcal{D} \to \mathbb{R}^+$ is $L$-*smooth* if for all $\mathbf{x}, \mathbf{y} \in \mathcal{D}$, $\|\nabla F(\mathbf{x}) - \nabla F(\mathbf{y})\| \leq L\|\mathbf{x} - \mathbf{y}\|$. Given a continuous monotone function $f : \mathcal{K} \to \mathbb{R}$, its *curvature* is defined as the smallest number $c \in [0, 1]$ such that $f(\mathbf{y} + \mathbf{z}) - f(\mathbf{y}) \geq (1 - c)(f(\mathbf{x} + \mathbf{z}) - f(\mathbf{x}))$, for all $\mathbf{x}, \mathbf{y} \in \mathcal{K}$ and $\mathbf{z} \geq 0$ such that $\mathbf{x} + \mathbf{z}, \mathbf{y} + \mathbf{z} \in \mathcal{K}$. We define the curvature of a function class $\mathbf{F}$ as the supremum of the curvature of functions in $\mathbf{F}$.

We say $\tilde{\nabla} f : \mathcal{K} \to \mathbb{R}^d$ is a $\mu$-*strongly $\gamma$-weakly up-super-gradient* of $f$ if for all $\mathbf{x} \leq \mathbf{y}$ in $\mathcal{K}$, we have $\gamma(\langle \tilde{\nabla} f(\mathbf{y}), \mathbf{y} - \mathbf{x} \rangle + \frac{\mu}{2} \|\mathbf{y} - \mathbf{x}\|^2) \leq f(\mathbf{y}) - f(\mathbf{x}) \leq \frac{1}{\gamma}(\langle \tilde{\nabla} f(\mathbf{x}), \mathbf{y} - \mathbf{x} \rangle - \frac{\mu}{2} \|\mathbf{y} - \mathbf{x}\|^2)$. Then we say $f$ is $\mu$-strongly $\gamma$-weakly up-concave if it is continuous and it has a $\mu$-strongly up-super-gradient. When $\gamma = 1$ and the above inequality holds for all $\mathbf{x}, \mathbf{y} \in \mathcal{K}$, we say $f$ is $\mu$-strongly concave. A differentiable function $f : \mathcal{K} \to \mathbb{R}$ is called *continuous DR-submodular* if for all $\mathbf{x} \leq \mathbf{y}$, we have $\nabla f(\mathbf{x}) \geq \nabla f(\mathbf{y})$. More generally, we say $f$ is $\gamma$-*weakly continuous DR-submodular* if for all $\mathbf{x} \leq \mathbf{y}$, we have $\nabla f(\mathbf{x}) \geq \gamma \nabla f(\mathbf{y})$. It follows that any $\gamma$-weakly continuous DR-submodular functions is $\gamma$-weakly up-concave.

## 3 PROBLEM SETUP

Online optimization problems can be formalized as a repeated game between an agent and an adversary. The game lasts for $T$ rounds on a convex domain $\mathcal{K}$ where $T$ and $\mathcal{K}$ are known to both players. In the $t$-th round, the agent chooses an action $\mathbf{x}_t$ from an action set $\mathcal{K} \subseteq \mathbb{R}^d$, then the adversary chooses a loss function $f_t \in \mathbf{F}$ and a query oracle for the function $f_t$. Then, for $1 \leq i \leq k_t$, the agent chooses a points $\mathbf{y}_{t,i}$ and receives the output of the query oracle.

To be more precise, an agent consists of a tuple $(\Omega^{\mathcal{A}}, \mathcal{A}^{\text{action}}, \mathcal{A}^{\text{query}})$, where $\Omega^{\mathcal{A}}$ is a probability space that captures all the randomness of $\mathcal{A}$. We assume that, before the first action, the agent

samples $\omega \in \Omega$. The next element in the tuple, $\mathcal{A}^{\mathrm{action}} = (\mathcal{A}_1^{\mathrm{action}}, \cdots, \mathcal{A}_T^{\mathrm{action}})$ is a sequence of functions such that $\mathcal{A}_t$ that maps the history $\Omega^{\mathcal{A}} \times \mathcal{K}^{t-1} \times \prod_{s=1}^{t-1} (\mathcal{K} \times \mathcal{O})^{k_s}$ to $\mathbf{x}_t \in \mathcal{K}$ where we use $\mathcal{O}$ to denote range of the query oracle. The last element in the tuple, $\mathcal{A}^{\mathrm{query}}$, is the query policy. For each $1 \le t \le T$ and $1 \le i \le k_t$, $\mathcal{A}_{t,i}^{\mathrm{query}} : \Omega^{\mathcal{A}} \times \mathcal{K}^t \times \prod_{s=1}^{t-1} (\mathcal{K} \times \mathcal{O})^{k_s} \times (\mathcal{K} \times \mathcal{O})^{i-1}$ is a function that, given previous actions and observations, either selects a point $\mathbf{y}_t^i \in \mathcal{K}$, i.e., query, or signals that the query policy at this time-step is terminated. We may drop $\omega$ as one of the inputs of the above functions when there is no ambiguity. We say the agent query function is *trivial* if $k_t = 1$ and $\mathbf{y}_{t,1} = \mathbf{x}_t$ for all $1 \le t \le T$. In this case, we simplify the notation and use the notation $\mathcal{A} = \mathcal{A}^{\mathrm{action}} = (\mathcal{A}_1, \cdots, \mathcal{A}_T)$ to denote the agent action functions and assume that the domain of $\mathcal{A}_t$ is $\Omega^{\mathcal{A}} \times (\mathcal{K} \times \mathcal{O})^{t-1}$.

A query oracle is a function that provides the observation to the agent. Formally, a query oracle for a function $f$ is a map $\mathcal{Q}$ defined on $\mathcal{K}$ such that for each $\mathbf{x} \in \mathcal{K}$, the $\mathcal{Q}(\mathbf{x})$ is a random variable taking value in the observation space $\mathcal{O}$. The query oracle is called a *stochastic value oracle* or *stochastic zeroth order oracle* if $\mathcal{O} = \mathbb{R}$ and $f(\mathbf{x}) = \mathbb{E}[\mathcal{Q}(\mathbf{x})]$. Similarly, it is called a *stochastic up-super-gradient oracle* or *stochastic first order oracle* if $\mathcal{O} = \mathbb{R}^d$ and $\mathbb{E}[\mathcal{Q}(\mathbf{x})]$ is a up-super-gradient of $f$ at $\mathbf{x}$. In all cases, if the random variable takes a single value with probability one, we refer to it as a *deterministic* oracle. Note that, given a function, there is at most a single deterministic gradient oracle, but there may be many deterministic up-super-gradient oracles. We will use $\nabla$ to denote the deterministic gradient oracle. We say an oracle is bounded by $B$ if its output is always within the Euclidean ball of radius $B$ centered at the origin. We say the agent takes *semi-bandit feedback* if the oracle is first-order and the agent query function is trivial. Similarly, it takes *bandit feedback* if the oracle is zeroth-order and the agent query function is trivial[1]. If the agent query function is non-trivial, then we say the agent requires *full-information feedback*.

An adversary Adv is a set such that each element $\mathcal{B} \in \mathrm{Adv}$, referred to as a *realized adversary*, is a sequence $(\mathcal{B}_1, \cdots, \mathcal{B}_T)$ of functions where each $\mathcal{B}_t$ maps a tuple $(\mathbf{x}_1, \cdots, \mathbf{x}_t) \in \mathcal{K}^t$ to a tuple $(f_t, \mathcal{Q}_t)$ where $f_t \in \mathbf{F}$ and $\mathcal{Q}_t$ is a query oracle for $f_t$. We say an adversary Adv is *oblivious* if for any realization $\mathcal{B} = (\mathcal{B}_1, \cdots, \mathcal{B}_T)$, all functions $\mathcal{B}_t$ are constant, i.e., they are independent of $(\mathbf{x}_1, \cdots, \mathbf{x}_t)$. In this case, a realized adversary may be simply represented by a sequence of functions $(f_1, \cdots, f_T) \in \mathbf{F}^T$ and a sequence of query oracles $(\mathcal{Q}_1, \cdots, \mathcal{Q}_T)$ for these functions. We say an adversary is a *weakly adaptive* adversary if each function $\mathcal{B}_t$ described above does not depend on $\mathbf{x}_t$ and therefore may be represented as a map defined on $\mathcal{K}^{t-1}$. In this work we also consider adversaries that are *fully adaptive*, i.e., adversaries with no restriction. Clearly any oblivious adversary is a weakly adaptive adversary and any weakly adaptive adversary is a fully adaptive adversary. Given a function class $\mathbf{F}$ and $i \in \{0, 1\}$, we use $\mathrm{Adv}_i^{\mathrm{f}}(\mathbf{F})$ to denote the set of all possible realized adversaries with deterministic $i$-th order oracles. If the oracle is instead stochastic and bounded by $B$, we use $\mathrm{Adv}_i^{\mathrm{f}}(\mathbf{F}, B)$ to denote such an adversary. Finally, we use $\mathrm{Adv}_i^{\mathrm{o}}(\mathbf{F})$ and $\mathrm{Adv}_i^{\mathrm{o}}(\mathbf{F}, B)$ to denote all oblivious realized adversaries with $i$-th order deterministic and stochastic oracles, respectively.

In order to handle different notions of regret with the same approach, for an agent $\mathcal{A}$, adversary Adv, compact set $\mathcal{U} \subseteq \mathcal{K}^T$, approximation coefficient $0 < \alpha \le 1$ and $1 \le a \le b \le T$, we define *regret* as

$$\mathcal{R}_{\alpha, \mathrm{Adv}}^{\mathcal{A}}(\mathcal{U})[a, b] := \sup_{\mathcal{B} \in \mathrm{Adv}} \mathbb{E}\left[ \alpha \max_{\mathbf{u} = (\mathbf{u}_1, \cdots, \mathbf{u}_T) \in \mathcal{U}} \sum_{t=a}^{b} f_t(\mathbf{u}_t) - \sum_{t=a}^{b} f_t(\mathbf{x}_t) \right],$$

where the expectation in the definition of the regret is over the randomness of the algorithm and the query oracle. We use the notation $\mathcal{R}_{\alpha, \mathcal{B}}^{\mathcal{A}}(\mathcal{U})[a, b] := \mathcal{R}_{\alpha, \mathrm{Adv}}^{\mathcal{A}}(\mathcal{U})[a, b]$ when $\mathrm{Adv} = \{\mathcal{B}\}$ is a singleton. We may drop $\alpha$ when it is equal to 1. When $\alpha < 1$, we often assume that the functions are non-negative. *Static adversarial regret* or simply *adversarial regret* corresponds to $a = 1$, $b = T$ and $\mathcal{U} = \mathcal{K}_\star^T := \{(\mathbf{x}, \cdots, \mathbf{x}) \mid \mathbf{x} \in \mathcal{K}\}$. When $a = 1$, $b = T$ and $\mathcal{U}$ contains only a single element then it is referred to as the *dynamic regret* (Zinkevich, 2003; Zhang et al., 2018). *Adaptive regret*, is defined as $\max_{1 \le a \le b \le T} \mathcal{R}_{\alpha, \mathrm{Adv}}^{\mathcal{A}}(\mathcal{K}_\star^T)[a, b]$ (Hazan & Seshadri, 2009). We drop $a$, $b$ and $\mathcal{U}$ when the statement is independent of their value or their value is clear from the context.

---

[1]This is a slight generalization of the common use of the term bandit feedback. Usually, bandit feedback refers to the case where the oracle is a *deterministic* zeroth-order oracle and the agent query function is trivial.

## 4 UNIFORM WRAPPERS

We next introduce a class of meta-algorithms that will be a central element of our proposed framework for adapting algorithms. At a high level, the meta-algorithms we consider wrap around the base algorithm and translate each action and feedback signal between the base algorithm and the adversary. The qualifier "uniform" highlights that the translations are one-to-one and independent of time.

**Definition 1.** Given a function class $\mathbf{F}$ and a family of query oracles $\mathcal{Q}$ over $\mathbf{F}$, we say a *uniform wrapper* $\mathcal{W} = (\mathcal{W}^{\text{action}}, \mathcal{W}^{\text{function}}, \mathcal{W}^{\text{query}})$ is a tuple of maps where $\mathcal{W}^{\text{action}} : \mathcal{K} \to \mathcal{K}$, $\mathcal{W}^{\text{function}} : \mathbf{F} \to \mathbf{H}$ for a function class $\mathbf{H}$ and for any $f \in \mathbf{F}$ and any query oracle $\mathcal{Q}_f \in \mathcal{Q}$, $\mathcal{W}^{\text{query}}(\mathcal{Q}_f)$ is a query oracle for $\mathcal{W}^{\text{function}}(f) \in \mathbf{H}$. Given an adversary $\text{Adv}$ choosing functions in $\mathbf{F}$ and query oracles in $\mathcal{Q}$, we define $\mathcal{W}(\text{Adv})$ to be the adversary over $\mathbf{H}$ where the selected function and query by the adversary are transformed according to $\mathcal{W}^{\text{function}}$ and $\mathcal{W}^{\text{query}}$. We say $\mathcal{W} = \text{Id}$ if all the maps in $\mathcal{W}$ are identity.

In Section 7 we will discuss several examples of uniform wrappers for up-concave optimization. We drop the superscripts and use $\mathcal{W}(\mathbf{x})$, $\mathcal{W}(f)$ and $\mathcal{W}(\mathcal{Q}_f)$ to denote $\mathcal{W}^{\text{action}}(\mathbf{x})$, $\mathcal{W}^{\text{function}}(f)$ and $\mathcal{W}^{\text{query}}(\mathcal{Q}_f)$, respectively, when there is no ambiguity.

Meta-algorithm 1 details the pseudo-code for $\mathcal{W}(\mathcal{A})$ for a uniform wrapper $\mathcal{W}$ and an online optimization algorithm $\mathcal{A}$. Note that, when $\mathcal{W} = \text{Id}$, the meta-algorithm also reduces to the identity meta-algorithm and we see that $\mathcal{W}(\mathcal{A}) = \mathcal{A}$. Note that in the special case where $\mathcal{A}$ is an online algorithm with semi-bandit feedback, Meta-algorithm 1 reduces to Algorithm 1 in (Pedramfar & Aggarwal, 2024a).

---

**Meta-algorithm 1:** Application of a uniform wrapper to the base algorithm - $\mathcal{W}(\mathcal{A})$

---

**Input :** horizon $T$, algorithm $\mathcal{A}$, uniform wrapper $\mathcal{W}$

**for** $t = 1, 2, \ldots, T$ **do**
 Play $\mathcal{W}^{\text{action}}(\mathbf{x}_t)$ where $\mathbf{x}_t$ is the action chosen by $\mathcal{A}^{\text{action}}$
 The adversary selects $f_t$ and a query oracle $\mathcal{Q}_t$ for $f_t$
 **for** $i$ *starting from 1, while* $\mathcal{A}^{query}$ *is not terminated for this time-step* **do**
  Let $\mathbf{y}_{t,i}$ be the query chosen by $\mathcal{A}^{\text{query}}$
  Return $\mathbf{o}_{t,i} = \mathcal{W}^{\text{query}}(\mathcal{Q}_t)(\mathbf{y}_{t,i})$ as the output of the query oracle to $\mathcal{A}^{\text{query}}$
 **end**
**end**

---

*In this paper, we will design uniform wrappers that could convert algorithms for concave optimization into algorithms for more general class of functions that contains many DR-submodular functions. Specifically, we consider upper-quadratizable/linearizable functions which we will discuss in the following section.*

## 5 LINEARIZABLE AND QUADRATIZABLE FUNCTIONS CLASSES

We next define an important function class significantly generalizes concavity but preserves enough structure that will enable us to obtain improved regret bounds for various problems.

**Definition 2** ((Pedramfar & Aggarwal, 2024a)). Let $\mathcal{K} \subseteq \mathbb{R}^d$ be a convex set, $\mathbf{F}$ be a function class over $\mathcal{K}$. We say the function class $\mathbf{F}$ is *upper quadratizable* if there are maps $\mathfrak{g} : \mathbf{F} \times \mathcal{K} \to \mathbb{R}^d$ and $h : \mathcal{K} \to \mathcal{K}$ and constants $\mu \geq 0$, $0 < \alpha \leq 1$ and $\beta > 0$ such that

$$\alpha f(\mathbf{y}) - f(h(\mathbf{x})) \leq \beta \left( \langle \mathfrak{g}(f, \mathbf{x}), \mathbf{y} - \mathbf{x} \rangle - \frac{\mu}{2} \|\mathbf{y} - \mathbf{x}\|^2 \right). \tag{1}$$

As a special case, when $\mu = 0$, we say $\mathbf{F}$ is *upper linearizable*. By setting $\mathfrak{g}(f, \mathbf{x}) = \nabla f(\mathbf{x})$, $h = \text{Id}_{\mathcal{K}}$ and $\alpha = \beta = 1$, we see that the notion of upper linearizability generalizes concavity and upper quadratizability generalizes strong concavity. It was shown in (Pedramfar & Aggarwal, 2024a) that several classes of DR-submodular (and up-concave) functions are upper quadratizable. (see Lemmas 1, 2 and 3) A similar notion of *lower-quadratizable/linearizable* may be similarly defined for minimization problems such as convex minimization [2].

---

[2]We say $\mathbf{F}$ is lower quadratizable if $\alpha f(\mathbf{y}) - f(h(\mathbf{x})) \geq \beta \left( \langle \mathfrak{g}(f, \mathbf{x}), \mathbf{y} - \mathbf{x} \rangle + \frac{\mu}{2} \|\mathbf{y} - \mathbf{x}\|^2 \right)$. This generalizes the notion of convexity and strong convexity.

**Definition 3.** We say $\mathbf{F}$ is *upper quadratizable with a uniform wrapper* $\mathcal{W}$ if $\mathcal{W}(\mathbf{F})$ is defined and differentiable over $\mathcal{K}$ and, for all $f \in \mathbf{F}$, we have

$$\alpha f(\mathbf{y}) - f(\mathcal{W}(\mathbf{x})) \leq \beta \left( \langle \nabla \mathcal{W}(f)(\mathbf{x}), \mathbf{y} - \mathbf{x} \rangle - \frac{\mu}{2} \|\mathbf{y} - \mathbf{x}\|^2 \right). \tag{2}$$

Note that a uniform wrapper is not uniquely determined by $h$ and $\mathfrak{g}$ in the definition of upper quadratizable functions as it also needs to describe transformations of query oracles. The special case with $\alpha = \beta = 1$, $\mathcal{W} = \mathrm{Id}$ reduces to the definition of (strong) concavity. In Section 7, we will construct uniform wrappers for several classes of upper quadratizable functions.

## 6    When Is Concave Optimization Enough?

As can be seen in Meta-algorithm 1, we may apply a uniform wrapper $\mathcal{W}$ to any online optimization algorithm $\mathcal{A}$. However, even if the original algorithm has a sublinear regret over concave functions and $\mathbf{F}$ is a function class that is upper quadratizable with $\mathcal{W}$, this does not guarantee that the resulting algorithm $\mathcal{W}(\mathcal{A})$ has a sublinear regret over $\mathbf{F}$. In this section we discuss how we might convert the proofs of the regret bound for $\mathcal{A}$ over concave functions into a proof of a similar regret bound for $\mathcal{W}(\mathcal{A})$ over $\mathbf{F}$. We will refer to algorithms $\mathcal{A}$ where the regret bounds could be be converted as *wrappable* algorithms.

The core idea for converting proof for concave optimization into proofs for upper-quadratizable optimization can be informally summarized in a few steps:

(0) Sometimes, if the algorithm $\mathcal{A}$ is the result of application of a meta-algorithm to another algorithm $\mathcal{B}$, e.g. $\mathcal{A} = \mathrm{SFTT}(\mathcal{B})$ (the meta-algorithm SFTT converts algorithms that require full-information feedback to ones that work with (semi-)bandit feedback; see Appendix J), we may need to consider the base algorithm instead. For example, in the example of SFTT, we might want to consider $\mathrm{SFTT}(\mathcal{W}(\mathcal{B}))$ instead of $\mathcal{W}(\mathrm{SFTT}(\mathcal{B})) = \mathcal{W}(\mathcal{A})$.

(1) Rewrite the parts of proof (after possibly adapting the algorithm) of the original regret bound without assuming that the function class in concave, in order to isolate the use on concavity in the proof. In this step, we hope to obtain a result that would only require a single use of an inequality of the type $f(\mathbf{y}) - f(\mathbf{x}) \leq \langle \nabla f(\mathbf{x}), \mathbf{y} - \mathbf{x} \rangle - \frac{\mu}{2} \|\mathbf{y} - \mathbf{x}\|^2$ to complete the proof for the concave case. See Theorems 1 (as an example of a family of zeroth order results) and 9 (as an example of a family of first order results) for examples of this step.

(2) Verify that the results of the previous step could be adapted to upper-quadratizable setting. See the proof of Theorems 2 and 10 for examples of this step.

In the following subsection, we discuss a version of Follow The Regularized Leader (FTRL) algorithm for concave optimization and adapt it to fit the guidelines discussed above. As another application of the guideline, we refer to Appendix B for a discussion of applying this guideline to recover some previous results in the literature, including all the results in Tables 1 and 2 that are marked with (*).

### 6.1    Follow The Regularized Leader

Follow The Regularized Leader is a popular online optimization algorithm. When applied to a sequence of vectors $\{\mathbf{g}_t\}_{t=1}^T$ in $\mathcal{K}$, FTRL outputs a sequence of points $\{\mathbf{x}_t\}_{t=1}^T$, where

$$\mathbf{x}_1 = \operatorname*{argmin}_{\mathbf{x} \in \mathcal{K}} \Phi(\mathbf{x}), \quad \mathbf{x}_{t+1} = \operatorname*{argmin}_{\mathbf{x} \in \mathcal{K}} \eta \sum_{s=1}^t \langle -\mathbf{g}_s, \mathbf{x} \rangle + \Phi(\mathbf{x}). \tag{3}$$

Here $\Phi(\mathbf{x})$ is an arbitrary regularizer and $\eta$ is a parameter. In this paper, we use a self-concordant barrier of $\mathcal{K}$ as the regularizer of FTRL. Self-concordant barriers were first proposed in the convex optimization literature, with (Abernethy et al., 2008) the first use in bandit feedback setting. We refer to Appendix E for an overview of the main ideas present in FTRL, including the definition of self-concordant barrier $\Phi$, the Minkowski set $\mathcal{K}_{\gamma, \mathbf{x}_1}$, and $\Sigma$-smoothing of function $f$ to obtain $f^\Sigma$.

Here we propose a FTRL variant for zeroth-order feedback, based on (Saha & Tewari, 2011), which will be a key base algorithm for our framework. See Algorithm 2 for pseudo-code.

The following theorems demonstrate how to apply the guideline described in the beginning of Section 6 to the results of (Saha & Tewari, 2011). The first step is to analyze the proof and modify the base algorithm so that we could obtain a result that is valid for non-convex functions and would only require a single use of an inequality similar to $f(\mathbf{y}) - f(\mathbf{x}) \leq \langle \nabla f(\mathbf{x}), \mathbf{y} - \mathbf{x} \rangle$ to obtain a regret bound for concave case. By a small modification in the original algorithm, we get ZO-FTRL which differs from the original in that it is no longer a bandit algorithm. While the agent plays $\mathbf{x}_t$ it queries the oracle at $\mathbf{x}_t + \delta \Sigma_t \mathbf{v}_t \neq \mathbf{x}_t$. This modification allows us to obtain the following result.

---

**Algorithm 2:** Zeroth Order Follow The Regularized Leader - ZO-FTRL

**Input :** Horizon $T$, smoothing radius $\delta$, learning rate $\eta$, $\nu$-self-concordant barrier $\Phi$

$\mathbf{x}_1 \leftarrow \arg\min_{\mathbf{x} \in \mathcal{K}} \Phi(\mathbf{x})$

**for** $t = 1, 2, \ldots, T$ **do**

    Play $\mathbf{x}_t$

    The adversary selects $f_t$ and reveals a zeroth-order query oracle $\mathcal{Q}_t$ for $f_t$

    $\Sigma_t \leftarrow \left( \nabla^2 \Phi(\mathbf{x}_t) \right)^{-1/2}$

    Draw $\mathbf{v}_t$ uniformly from $\mathbb{S}^{d-1}$

    $y_t \leftarrow$ a sample of $Q_t$ at $\mathbf{x}_t + \delta \Sigma_t \mathbf{v}_t$

    $\mathbf{o}_t \leftarrow \frac{d}{\delta} y_t \Sigma_t^{-1} \mathbf{v}_t$

    $\mathbf{x}_{t+1} \leftarrow \arg\min_{\mathbf{x} \in \mathcal{K}} \sum_{s=1}^t -\eta \langle \mathbf{o}_t, \mathbf{x} \rangle + \Phi(\mathbf{x})$

**end**

---

**Theorem 1.** *Let $\mathbf{F}$ be an $M_1$-Lipschitz $M_2$-smooth function class that is bounded by $M_0$ and let $B_0 \geq M_0$. Also let $\mathcal{B} \in \mathrm{Adv}_0^o(\mathbf{F}, B_0)$ be a realized adversary that returns $f_1, \cdots, f_T$, let $\mathbf{u}_* \in \arg\max_{\mathbf{u} \in \mathcal{K}} \sum_{t=1}^T f_t(\mathbf{u})$ and $\hat{\mathbf{u}}_* \in \arg\min_{\mathbf{x} \in \mathcal{K}_{\gamma, \mathbf{x}_1}} \|\mathbf{u}_* - \mathbf{x}\|$ where $\gamma = T^{-1}$. Then, when running Algorithm 2 against $\mathcal{B}$, we have*

$$\sum_{t=1}^T \mathbb{E}\left[ f_t(\mathbf{u}_*) - f_t(\mathbf{x}_t) \right] - O(\delta^2 T) \leq \sum_{t=1}^T \mathbb{E}\left[ f_t^{\delta \Sigma_t}(\hat{\mathbf{u}}_*) - f_t^{\delta \Sigma_t}(\mathbf{x}_t) \right],$$

*and* $$\sum_{t=1}^T \mathbb{E}\left[ \langle \nabla f_t^{\delta \Sigma_t}(\mathbf{x}_t), \hat{\mathbf{u}}_* - \mathbf{x}_t \rangle \right] \leq O\left( \eta \delta^{-2} T + \eta^{-1} \log T \right).$$

See Appendix G for the proof. Note that if $f$ is concave, then we use use Lemma 4 to see that the right hand side of the first inequality is bounded by the left hand side of the second inequality and obtain the regret bound for the concave case. See Appendix H for the proof.

**Theorem 2.** *Let $\mathbf{F}$ be an $M_1$-Lipschitz $M_2$-smooth function class over $\mathcal{K}$ that is upper-linearizable with $0 < \alpha \leq 1$, $\beta \geq 0$ and a zeroth-order uniform wrapper $\mathcal{W}$. Also assume that $\mathcal{W}^{action}$ is $M_1'$-Lipschitz and $M_2'$-smooth. If $\mathrm{Adv}$ is a zeroth order oblivious adversary over $\mathbf{F}$ such that for for any $f \in \mathbf{F}$ and any query oracle $\mathcal{Q}_f$ returned by $\mathrm{Adv}$, $\mathcal{W}(\mathcal{Q}_f)$ is a stochastic zeroth order query oracle for $\mathcal{W}(f)$ that is bounded by $B_0$, then*

$$\mathcal{R}_{\alpha, \mathrm{Adv}}^{\mathcal{W}(\text{ZO-FTRL})} = O\left( \eta \delta^{-2} T + \eta^{-1} \log T + \delta^2 T \right),$$

*In particular, by setting $\eta = T^{-2/3}$ and $\delta = T^{-1/6}$, we see that $\mathcal{R}_{\alpha, \mathrm{Adv}}^{\mathcal{W}(\text{ZO-FTRL})} = \tilde{O}(T^{2/3})$.*

## 7 Uniform wrappers for up-concave optimization

In this section, we study three classes of up-concave functions and show that they are upper-quadratizable with appropriate uniform wrappers. By identifying appropriate uniform wrappers, Theorem 2 immediately implies $\tilde{O}(T^{2/3})$ $\alpha$-regret using UNIFORMWRAPPER with ZO-FTRL as a base algorithm along with the respective uniform wrapper.

### 7.1 Monotone $\mu$-strongly $\gamma$-weakly up-concave functions with bounded curvature ($\mathbf{F}^M$)

For differentiable DR-submodular functions, the following lemma is proved for the case $\gamma = 1$ in (Fazel & Sadeghi, 2023) and for the case $\mu = 0$ in (Hassani et al., 2017). The general form we use here is proved in Lemma 1 in (Pedramfar & Aggarwal, 2024a).

**Lemma 1.** *Let $f : [0,1]^d \to \mathbb{R}$ be a non-negative monotone $\mu$-strongly $\gamma$-weakly up-concave function with curvature bounded by $c$. Then, for all $\mathbf{x}, \mathbf{y} \in [0,1]^d$, we have*

$$\frac{\gamma^2}{1+c\gamma^2} f(\mathbf{y}) - f(\mathbf{x}) \leq \frac{\gamma}{1+c\gamma^2} \big( \langle \tilde{\nabla} f(\mathbf{x}), \mathbf{y} - \mathbf{x} \rangle - \frac{\mu}{2} \|\mathbf{y} - \mathbf{x}\|^2 \big),$$

*where $\tilde{\nabla} f$ is an up-super-gradient for $f$.*

Lemma 1, together with Definition 1 of uniform wrappers, immediately imply the following.

**Theorem 3.** *Let $\mathbf{F}^{\mathrm{M}}$ be the class of functions over $\mathcal{K}$ where every $f \in \mathbf{F}^{\mathrm{M}}$ may be extended to a non-negative differentiable monotone $\mu$-strongly $\gamma$-weakly up-concave function with curvature bounded by $c$ defined over $[0,1]^d$. Then $\mathbf{F}^{\mathrm{M}}$ is upper-quadratizable with uniform wrapper $\mathcal{W}^{\mathrm{M}} = \mathrm{Id}$.*

If $\mathcal{A}$ is a wrappable algorithm for online optimization with sublinear regret bound of $O(T^\beta)$ for some $\beta < 1$ over concave functions, then the above theorem shows that by directly applying $\mathcal{A}$ to monotone DR-submodular functions, we get $\frac{\gamma}{1+c\gamma^2}$-regret bound of $O(T^\beta)$. As a special case, when $\mathcal{A}$ is one of the wrappable algorithm described in Theorem 10, using the above theorem recovers Theorem 2 in (Pedramfar & Aggarwal, 2024a) which itself is a generalization of Theorem 2 in (Chen et al., 2018) and Theorem 3 in (Fazel & Sadeghi, 2023).

## 7.2 Monotone $\gamma$-weakly up-concave functions over convex sets containing the origin $(\mathbf{F}^{\mathrm{M0}})$

For differentiable monotone DR-submodular functions, the following lemma is proved in (Zhang et al., 2022). The general form here is proved in Lemma 2 in (Pedramfar & Aggarwal, 2024a).

**Lemma 2.** *Let $f : [0,1]^d \to \mathbb{R}$ be a non-negative monotone $\gamma$-weakly up-concave differentiable function and let $F : [0,1]^d \to \mathbb{R}$ be the function defined by $F(\mathbf{x}) := \int_0^1 \frac{\gamma e^{\gamma(z-1)}}{(1-e^{-\gamma})z} (f(z * \mathbf{x}) - f(\mathbf{0})) dz$. Then $F$ is differentiable and*

$$(1 - e^{-\gamma}) f(\mathbf{y}) - f(\mathbf{x}) \leq \frac{1 - e^{-\gamma}}{\gamma} \langle \nabla F(\mathbf{x}), \mathbf{y} - \mathbf{x} \rangle.$$

*Let the random variable $\mathcal{Z}^{\mathrm{M0}} \in [0,1]$ be defined by the law $\forall z \in [0,1], \quad \mathbb{P}(\mathcal{Z}^{\mathrm{M0}} \leq z) = \int_0^z \frac{\gamma e^{\gamma(u-1)}}{1-e^{-\gamma}} du$. Then we have $\mathbb{E}_{z \sim \mathcal{Z}^{\mathrm{M0}}} \big[ z^{-1}(f(z * \mathbf{x}) - f(\mathbf{0})) \big] = F(\mathbf{x})$. Moreover, for $i \geq 1$, if $f$ is $i$ times differentiable then we also have $\mathbb{E}_{z \sim \mathcal{Z}^{\mathrm{M0}}} \big[ z^{i-1} \nabla^i f(z * \mathbf{x}) \big] = \nabla^i F(\mathbf{x})$.*

**Definition 4.** Let $\mathcal{K} \subseteq [0,1]^d$ be a convex set containing the origin and, for any $i \geq 0$, let $\mathbf{F}_i^{\mathrm{M0}}$ be the class of functions over $\mathcal{K}$ that are $\max\{i, 1\}$ times differentiable and where every $f \in \mathbf{F}_i^{\mathrm{M0}}$ may be extended to a non-negative monotone $\gamma$-weakly up-concave function defined over $[0,1]^d$. We also assume that $f(\mathbf{0}) = 0$ for all $f \in \mathbf{F}_0^{\mathrm{M0}}$. We define $\mathcal{W}_i^{\mathrm{M0}} := ((\mathcal{W}_i^{\mathrm{M0}})^{\mathrm{action}}, (\mathcal{W}_i^{\mathrm{M0}})^{\mathrm{function}}, (\mathcal{W}_i^{\mathrm{M0}})^{\mathrm{query}})$ to be the uniform wrapper with (i) $(\mathcal{W}_i^{\mathrm{M0}})^{\mathrm{action}} := \mathrm{Id}_{\mathcal{K}}$; (ii) for any $f \in \mathbf{F}_i^{\mathrm{M0}}$, $(\mathcal{W}_i^{\mathrm{M0}})^{\mathrm{function}}(f) := \mathbf{x} \mapsto \mathbb{E}_{z \sim \mathcal{Z}^{\mathrm{M0}}} \big[ z^{-1}(f(z * \mathbf{x}) - f(\mathbf{0})) \big] : \mathcal{K} \to \mathbb{R}$; and
(iii) for any $f \in \mathbf{F}_i^{\mathrm{M0}}$ and any $i$-th order query oracle $\mathcal{Q}_f$ for $f$, we have $(\mathcal{W}_i^{\mathrm{M0}})^{\mathrm{query}}(\mathcal{Q}_f)(\mathbf{x}) := z^{i-1} * Q_f(z * \mathbf{x})$, where $z$ is sampled according to $\mathbb{P}(\mathcal{Z}^{\mathrm{M0}} \leq z)$.

**Theorem 4.** *For any $i \geq 0$, the function class $\mathbf{F}_i^{\mathrm{M0}}$ defined above is upper-linearizable with the uniform wrapper $\mathcal{W}_i^{\mathrm{M0}}$.*

*Remark* 1. The meta-algorithm $\mathcal{A} \mapsto \mathrm{OMBQ}(\mathcal{A}, \mathrm{BQM0}, \mathrm{Id})$, described in (Pedramfar & Aggarwal, 2024a), is identical to $\mathcal{A} \mapsto \mathcal{W}_1^{\mathrm{M0}}(\mathcal{A})$. In other words, the results of Theorem 3 in (Pedramfar & Aggarwal, 2024a) are about the first order uniform wrapper $\mathcal{W}_1^{\mathrm{M0}}$. Here we consider a more general case where we are not necessarily limited to first order.

## 7.3 Non-monotone up-concave functions over general convex sets $(\mathbf{F}^{\mathrm{NM}})$

For differentiable monotone DR-submodular functions, the following lemma is proved in (Zhang et al., 2024). The general form we use is proven in Lemma 3 in (Pedramfar & Aggarwal, 2024a).

**Lemma 3.** *Let $f : [0,1]^d \to \mathbb{R}$ be a non-negative continuous up-concave differentiable function and let $\underline{\mathbf{x}} \in \mathcal{K}$. Define $F : [0,1]^d \to \mathbb{R}$ as the function $F(\mathbf{x}) := \int_0^1 \frac{2}{3z(1-\frac{z}{2})^3}(f(\frac{z}{2} * (\mathbf{x} - \underline{\mathbf{x}}) + \underline{\mathbf{x}}) - f(\underline{\mathbf{x}}))dz$, then $F$ is differentiable and we have*

$$\frac{1 - \|\underline{\mathbf{x}}\|_\infty}{4} f(\mathbf{y}) - f\left(\frac{\mathbf{x} + \underline{\mathbf{x}}}{2}\right) \le \frac{3}{8}\langle \nabla F(\mathbf{x}), \mathbf{y} - \mathbf{x} \rangle.$$

*Let the random variable $\mathcal{Z}^{\mathrm{NM}} \in [0,1]$ be defined by the law $\forall z \in [0,1]$, $\mathbb{P}(\mathcal{Z}^{\mathrm{NM}} \le z) = \int_0^z \frac{1}{3(1-\frac{u}{2})^3} du$. Then we have $\mathbb{E}_{z \sim \mathcal{Z}^{\mathrm{NM}}}[(\frac{z}{2})^{-1} * (f(\frac{z}{2} * (\mathbf{x} - \underline{\mathbf{x}}) + \underline{\mathbf{x}}) - f(\underline{\mathbf{x}}))] = F(\mathbf{x})$. Moreover, if $i \ge 1$ and $f$ is $i$ times differentiable, then $\mathbb{E}_{z \sim \mathcal{Z}^{\mathrm{NM}}}[(\frac{z}{2})^{i-1} * \nabla^i f(\frac{z}{2} * (\mathbf{x} - \underline{\mathbf{x}}) + \underline{\mathbf{x}})] = \nabla^i F(\mathbf{x})$.*

**Definition 5.** *Let $\mathcal{K} \subseteq [0,1]^d$ be a convex set and, for any $i \ge 0$, let $\mathbf{F}_i^{\mathrm{NM}}$ be the class of functions over $\mathcal{K}$ where every $f \in \mathbf{F}_i^{\mathrm{NM}}$ may be extended to a non-negative up-concave function defined over $[0,1]^d$. We also assume that $\mathbf{F}_i^{\mathrm{NM}}$ is $\max\{i,1\}$ times differentiable for all $i \ge 0$ and, for some known constant $c \ge 0$ and all $f \in \mathbf{F}_0^{\mathrm{NM}}$, $f(\underline{\mathbf{x}}) = c$. For $i \ge 0$, we define $\mathcal{W}_i^{\mathrm{NM}} = ((\mathcal{W}_i^{\mathrm{NM}})^{\mathrm{action}}, (\mathcal{W}_i^{\mathrm{NM}})^{\mathrm{function}}, (\mathcal{W}_i^{\mathrm{NM}})^{\mathrm{query}})$ to be the uniform wrapper with*
*(i) $(\mathcal{W}_i^{\mathrm{NM}})^{\mathrm{action}} := \mathbf{x} \mapsto \frac{\mathbf{x} + \underline{\mathbf{x}}}{2} : \mathcal{K} \to \mathcal{K}$; (ii) for any $f \in \mathbf{F}_i^{\mathrm{NM}}$,*
*$(\mathcal{W}_i^{\mathrm{NM}})^{\mathrm{function}}(f) := \mathbf{x} \mapsto \mathbb{E}_{z \sim \mathcal{Z}^{\mathrm{NM}}}[(\frac{z}{2})^{-1} * (f(\frac{z}{2} * (\mathbf{x} - \underline{\mathbf{x}}) + \underline{\mathbf{x}}) - f(\underline{\mathbf{x}}))] : \mathcal{K} \to \mathbb{R}$; and*
*(iii) for any $f \in \mathbf{F}_i^{\mathrm{NM}}$ and any $i$-th order query oracle $\mathcal{Q}_f$ for $f$,*

$$(\mathcal{W}_i^{\mathrm{NM}})^{\mathrm{query}}(\mathcal{Q}_f)(\mathbf{x}) := \begin{cases} \left(\frac{z}{2}\right)^{i-1} * Q_f\left(\frac{z}{2} * (\mathbf{x} - \underline{\mathbf{x}}) + \underline{\mathbf{x}}\right) & \text{if } i \ge 1 \\ \left(\frac{z}{2}\right)^{-1} * \left(Q_f\left(\frac{z}{2} * (\mathbf{x} - \underline{\mathbf{x}}) + \underline{\mathbf{x}}\right) - c\right) & \text{if } i = 0 \end{cases}$$

*where $z$ is sampled according to $\mathbb{P}(\mathcal{Z}^{\mathrm{NM}} \le z)$.*

**Theorem 5.** *For any $i \ge 0$, the function class $\mathbf{F}_i^{\mathrm{NM}}$ defined above is upper-linearizable with the uniform wrapper $\mathcal{W}_i^{\mathrm{NM}}$.*

*Remark* 2. The meta-algorithm $\mathcal{A} \mapsto \mathrm{OMBQ}(\mathcal{A}, \mathrm{BQN}, \mathbf{x} \mapsto \frac{\mathbf{x} + \underline{\mathbf{x}}}{2})$, described in (Pedramfar & Aggarwal, 2024a), is identical to $\mathcal{A} \mapsto \mathcal{W}_1^{\mathrm{NM}}(\mathcal{A})$. In other words, the results of Theorem 4 in (Pedramfar & Aggarwal, 2024a) are about the first order uniform wrapper $\mathcal{W}_1^{\mathrm{NM}}$. Here we consider a more general case where we are not necessarily limited to first order.

## 8 APPLICATIONS

We next discuss some specific online and offline non-convex/non-concave optimization problems for which we can use our new framework to derive improved regret and sample complexity bounds respectively by applying uniform wrappers proposed in Section 7 to the zeroth order feedback OCO base algorithm ZO-FTRL (Algorithm 2). We note that we can also apply our framework to other base algorithms to recover many existing results in the literature. (See Appendix B for more details).

We start with a definition. For $\mathbf{x} \in \mathcal{K}$ and $C > 0$, we say a zeroth order query oracle $\mathcal{Q}_f$ is contained in a $(\mathbf{x}, C)$ cone if we have $|Q_f(\mathbf{z}) - f(\mathbf{x})| \le C\|\mathbf{z} - \mathbf{x}\|$ for all $\mathbf{z} \in \mathcal{K}$. In other words, the randomness of the query oracle approaches to zero at least linearly as we approach the point $\mathbf{x}$. We use the notation $\mathrm{Adv}_0^o(\mathbf{F}, \mathrm{Cone}(\mathbf{x}, C))$ to denote the oblivious adversary over $\mathbf{F}$ with query oracles that are contained within this cone. Note that $\mathcal{Q}_f \in \mathrm{Adv}_0^o(\mathbf{F}, \mathrm{Cone}(\mathbf{x}, C))$ is equivalent to $\mathcal{W}_0^{\mathrm{NM}}(\mathcal{Q}_f)$ being bounded. See condition (iii) of Definition 5 for details. If $\mathcal{Q}_f$ does not belong to a cone as described above, we can see that the term $\left(\frac{z}{2}\right)^{-1}$ causes $\mathcal{W}_0^{\mathrm{NM}}(\mathcal{Q}_f)$ to blow up. Similarly, in the special case when $\underline{x} = \mathbf{0}$ and $f(\mathbf{0}) = 0$, it is also equivalent to $\mathcal{W}_0^{\mathrm{M0}}(\mathcal{Q}_f)$ being bounded.

We begin by showing $\tilde{O}(T^{2/3})$ $\alpha$-regret bounds for online optimization problems for the three function classes discussed in Section 7 under zeroth order feedback. See Appendix I for the proof.

**Theorem 6.** *Let $\mathbf{F}_0^{\mathrm{M}}$, $\mathbf{F}_0^{\mathrm{M0}}$ and $\mathbf{F}_0^{\mathrm{NM}}$ denote the function classes described in Lemmas 1, 2 and 3 respectively and let $\alpha^{\mathrm{M}}$, $\alpha^{\mathrm{M0}}$ and $\alpha^{\mathrm{NM}}$ be the values of $\alpha$. If the function classes are $M_1$-Lipschitz and $M_2$-smooth, then for any $C > 0$ and $B_0 \ge M_0 = \max_{\mathbf{x} \in \mathcal{K}} f(\mathbf{x})$, the following are $\tilde{O}(T^{2/3})$:*

$$\mathcal{R}_{\alpha^{\mathrm{M}}, \mathrm{Adv}_0^o(\mathbf{F}_0^{\mathrm{M}}, B_0)}^{\mathcal{W}_0^{\mathrm{M}}(\mathrm{ZO\text{-}FTRL})}, \quad \mathcal{R}_{\alpha^{\mathrm{M0}}, \mathrm{Adv}_0^o(\mathbf{F}_0^{\mathrm{M0}}, \mathrm{Cone}(\mathbf{0}, C))}^{\mathcal{W}_0^{\mathrm{M0}}(\mathrm{ZO\text{-}FTRL})}, \quad \mathcal{R}_{\alpha^{\mathrm{NM}}, \mathrm{Adv}_0^o(\mathbf{F}_0^{\mathrm{NM}}, \mathrm{Cone}(\underline{\mathbf{x}}, C))}^{\mathcal{W}_0^{\mathrm{NM}}(\mathrm{ZO\text{-}FTRL})}.$$

*Remark* 3. For each function class, the SOTA for noisy zeroth order feedback achieved $\tilde{O}(T^{3/4})$ $\alpha$-regret bounds while we achieve $\tilde{O}(T^{2/3})$. For the special case of exact zeroth order feedback, the SOTA is $\tilde{O}(\sqrt{T})$. All the SOTA algorithms mentioned are special cases of our framework.

We next show $\tilde{O}(T^{3/4})$ $\alpha$-regret bounds for online optimization problems for the three function classes discussed in Section 7 under bandit feedback. For full information zeroth order algorithms, the query location may differ from the action taken. Here we convert them into bandit algorithms using the meta-algorithm Stochastic Full-information To Trivial query (SFTT) from (Pedramfar & Aggarwal, 2024a) (see Appendix J for details). The proof is in Appendix K.

**Theorem 7.** *Under the assumptions of Theorem 6, the following are* $\tilde{O}(T^{3/4})$:

$$\mathcal{R}^{\text{SFTT}(\mathcal{W}_0^{\text{M}}(\text{ZO-FTRL}))}_{\alpha^{\text{M}},\text{Adv}_0^o(\mathbf{F}_0^{\text{M}},B_0)}, \quad \mathcal{R}^{\text{SFTT}(\mathcal{W}_0^{\text{M0}}(\text{ZO-FTRL}))}_{\alpha^{\text{M0}},\text{Adv}_0^o(\mathbf{F}_0^{\text{M0}},\text{Cone}(\mathbf{0},C))}, \quad \mathcal{R}^{\text{SFTT}(\mathcal{W}_0^{\text{NM}}(\text{ZO-FTRL}))}_{\alpha^{\text{NM}},\text{Adv}_0^o(\mathbf{F}_0^{\text{NM}},\text{Cone}(\underline{\mathbf{x}},C))},$$

*where* SFTT *is Algorithm 4 in (Pedramfar & Aggarwal, 2024a) with* $L = T^{1/4}$.

*Remark* 4. Note that Algorithm 3 in (Wan et al., 2023) is in fact $\text{SFTT}(\mathcal{W}_0^{\text{M0}}(\text{ZO-FTRL}))$. However, our analysis simplifies the proof and generalizes the result to allow for stochastic feedback.

*Remark* 5. For the class $\mathbf{F}^{\text{NM}}$ of non-monotone up-concave functions over general convex sets, our $\tilde{O}(T^{3/4})$ bound beats the SOTA $\tilde{O}(T^{4/5})$ bounds for exact and for noisy bandit feedback. For the class $\mathbf{F}^{\text{M0}}$ of monotone $\gamma$-weakly up-concave functions over convex sets containing the origin, our $\tilde{O}(T^{3/4})$ bound beats the SOTA $\tilde{O}(T^{4/5})$ bound for noisy bandit feedback and matches the bound for exact bandit feedback. For the third class $\mathbf{F}^{\text{M}}$ of monotone $\mu$-strongly $\gamma$-weakly up-concave functions with bounded curvature, our results match the SOTA. All of the SOTA algorithms mentioned here are special cases of our framework.

Conversions of online algorithms to offline are referred to online-to-batch techniques and are well-known in the literature (See (Shalev-Shwartz, 2012)). A simple approach is to simply run the online algorithm and if the actions chosen by the algorithm are $\mathbf{x}_1, \cdots, \mathbf{x}_T$, return $\mathbf{x}_t$ for $1 \leq t \leq T$ with probability $1/T$. We use OTB to denote the meta-algorithm that uses this approach to convert online algorithms to offline algorithms.

We next show that using OTB conversion (on top of $\mathcal{W}(\text{ZO-FTRL})$), we obtain $\tilde{O}(1/\epsilon^3)$ sample complexity for finding an $\alpha$-approximate solution in each function class under a noisy value oracle model, beating the SOTA $\tilde{O}(1/\epsilon^4)$ sample complexity. The proof is in Appendix L

**Theorem 8.** *Under the assumptions of Theorem 6, the following is true.*

*(i) If the stochastic query oracle is bounded by* $B_0$, *then the sample complexity of the offline algorithm* $\text{OTB}(\mathcal{W}_0^{\text{M}}(\text{ZO-FTRL}))$ *over* $\mathbf{F}_0^{\text{M}}$ *is* $\tilde{O}(\epsilon^{-3})$.

*(ii) If the stochastic query oracle is contained in the cone* $\text{Cone}(\mathbf{0}, C)$, *then the sample complexity of the offline algorithm* $\text{OTB}(\mathcal{W}_0^{\text{M0}}(\text{ZO-FTRL}))$ *over* $\mathbf{F}_0^{\text{M0}}$ *is* $\tilde{O}(\epsilon^{-3})$.

*(iii) If the stochastic query oracle is contained in the cone* $\text{Cone}(\underline{\mathbf{x}}, C)$, *then the sample complexity of the offline algorithm* $\text{OTB}(\mathcal{W}_0^{\text{NM}}(\text{ZO-FTRL}))$ *over* $\mathbf{F}_0^{\text{NM}}$ *is* $\tilde{O}(\epsilon^{-3})$.

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
