|---|---|---|---|---|---|---|
| Monotone | $0 \in \mathcal{K}$ | $\nabla F$ | stoch. | (Mokhtari et al., 2020) | $1-e^{-\gamma}$ | $O(1/\epsilon^3)$ |
| | | | | (Hassani et al., 2020) (*) | $1-e^{-\gamma}$ | $O(1/\epsilon^2)$ |
| | | | | (Zhang et al., 2022) (*) | $1-e^{-\gamma}$ | $O(1/\epsilon^2)$ |
| | | | | (Pedramfar & Aggarwal, 2024a) (*) | $1-e^{-\gamma}$ | $O(1/\epsilon^2)$ |
| | | $F$ | det. | (Pedramfar et al., 2024b) | $1-e^{-\gamma}$ | $O(1/\epsilon^3)$ |
| | | | | (Pedramfar & Aggarwal, 2024a) (*) | $1-e^{-\gamma}$ | $O(1/\epsilon^2)$ |
| | | | stoch. † | Theorem 8 | $1-e^{-\gamma}$ | $O(1/\epsilon^3)$ |
| | | | stoch. | (Pedramfar et al., 2024b) | $1-e^{-\gamma}$ | $O(1/\epsilon^5)$ |
| | | | | (Pedramfar & Aggarwal, 2024a) (*) | $1-e^{-\gamma}$ | $O(1/\epsilon^4)$ |
| | general | $\nabla F$ | stoch. | (Hassani et al., 2017) (*) | $\gamma^2/(1+\gamma^2)$ | $O(1/\epsilon^2)$ |
| | | | | (Pedramfar et al., 2024b) | $\gamma^2/(1+\gamma^2)$ | $\tilde{O}(1/\epsilon^3)$ |
| | | | | (Pedramfar & Aggarwal, 2024a) (*) | $\gamma^2/(1+c\gamma^2)$ | $O(1/\epsilon^2)$ |
| | | $F$ | det. | (Pedramfar et al., 2023) | $\gamma^2/(1+\gamma^2)$ | $O(1/\epsilon^3)$ |
| | | | | (Pedramfar & Aggarwal, 2024a) (*) | $\gamma^2/(1+c\gamma^2)$ | $O(1/\epsilon^2)$ |
| | | | stoch. | (Pedramfar et al., 2023) | $\gamma^2/(1+\gamma^2)$ | $\tilde{O}(1/\epsilon^5)$ |
| | | | | (Pedramfar & Aggarwal, 2024a) (*) | $\gamma^2/(1+c\gamma^2)$ | $O(1/\epsilon^4)$ |
| | | | | Theorem 8 | $\gamma^2/(1+c\gamma^2)$ | $\tilde{O}(1/\epsilon^3)$ |
| Non-Monotone | general | $\nabla F$ | stoch. | (Pedramfar et al., 2024b) | $\frac{\gamma(1-\gamma h)}{\gamma'-1}\left(\frac{1}{2}-\frac{1}{2^{\gamma'}}\right)$ | $O(1/\epsilon^3)$ |
| | | | | (Zhang et al., 2024) (*) | $(1-h)/4$ | $O(1/\epsilon^2)$ |
| | | | | (Pedramfar & Aggarwal, 2024a) (*) | $(1-h)/4$ | $O(1/\epsilon^2)$ |
| | | $F$ | det. | (Pedramfar et al., 2024b) | $\frac{\gamma(1-\gamma h)}{\gamma'-1}\left(\frac{1}{2}-\frac{1}{2^{\gamma'}}\right)$ | $O(1/\epsilon^3)$ |
| | | | | (Pedramfar & Aggarwal, 2024a) (*) | $(1-h)/4$ | $O(1/\epsilon^2)$ |
| | | | stoch. † | Theorem 8 | $(1-h)/4$ | $O(1/\epsilon^3)$ |
| | | | stoch. | (Pedramfar et al., 2024b) | $\frac{\gamma(1-\gamma h)}{\gamma'-1}\left(\frac{1}{2}-\frac{1}{2^{\gamma'}}\right)$ | $O(1/\epsilon^5)$ |
| | | | | (Pedramfar & Aggarwal, 2024a) (*) | $(1-h)/4$ | $O(1/\epsilon^4)$ |

This table compares the different results for the number of oracle calls (complexity) *within the constraint set* for up-concave maximization. We refer to (Pedramfar et al., 2024b) for a more comprehensive table that includes results for deterministic first order feedback. Here $h := \min_{\mathbf{z} \in \mathcal{K}} \|\mathbf{z}\|_\infty$ and $\gamma' := \gamma + 1/\gamma$. **Rows marked with (*) are results in the literature that fit within the framework described in this paper.** The rows describing results with stochastic feedback that are marked with † assume that the random query oracle is contained with a cone, as detailed in Theorem 6.

## A ADDITIONAL RELATED WORKS

**DR-submodular maximization** Two of the main methods for continuous DR-submodular maximization are *Frank-Wolfe type methods* and *Boosting based methods*. This division is based on how the approximation coefficient appears in the proof.

In Frank-Wolfe type algorithms, the approximation coefficient appears by specific choices of the Frank-Wolfe update rules. (See Lemma 8 in (Pedramfar et al., 2024a)) The specific choices of the update rules for different settings have been proposed in (Bian et al., 2017b;a; Mualem & Feldman, 2023; Pedramfar et al., 2023; Chen et al., 2023). The momentum technique of (Mokhtari et al., 2020) has been used to convert algorithms designed for deterministic feedback to stochastic feedback setting. (Hassani et al., 2020) proposed a Frank-Wolfe variant with access to a stochastic gradient oracle with *known distribution*. Frank-Wolfe type algorithms been adapted to the online setting using Meta-Frank-Wolfe (Chen et al., 2018; 2019) or using Blackwell approachablity (Niazadeh et al., 2023). Later (Zhang et al., 2019) used a Meta-Frank-Wolfe with random permutation technique to obtain full-information results that only require a single query per function and also bandit results. This was extended to another settings by (Zhang et al., 2023) and generalized to many different settings with improved regret bounds by (Pedramfar et al., 2024a).

Some techniques construct an alternative function such that maximization of this function results in approximate maximization of the original function. Given this definition, we may consider the result of (Hassani et al., 2017; Chen et al., 2018; Fazel & Sadeghi, 2023) as the first boosting based results. However, in the case of monotone DR-submodular functions over general convex sets, the alternative function is identical to the original function. The term boosting in this context was first used in (Zhang et al., 2022), based on ideas presented in (Filmus & Ward, 2012; Mitra et al., 2021), for monotone functions over convex sets containing the origin. This idea has been used later in (Wan et al., 2023; Liao et al., 2023) in bandit and projection-free full-information settings. Finally, in (Zhang et al., 2024) a boosting based method was introduced for non-monotone functions over general convex sets.

**Up-concave maximization** Not all continuous DR-submodular functions are concave and not all concave functions are continuous DR-submodular. (Mitra et al., 2021) considers functions that are

the sum of a concave and a continuous DR-submodular function. It is well-known that continuous DR-submodular functions are concave along positive directions (Calinescu et al., 2011; Bian et al., 2017b). Based on this idea, (Wilder, 2018) defined an up-concave function as a function that is concave along positive directions. Up-concave maximization has been considered in the offline setting before, e.g. in (Lee et al., 2023) and (Pedramfar & Aggarwal, 2024a). In this work, we focus on up-concave maximization which is a generalization of DR-submodular maximization.

## B   RECOVERING PREVIOUS RESULTS IN THE LITERATURE

As mentioned in Remark 4, Algorithm 3 in (Wan et al., 2023) is in fact $\text{SFTT}(\mathcal{W}_0^{\text{M0}}(\text{ZO-FTRL}))$ and therefore their result fits within our framework. The way the remaining results in the tables that are marked with (*) is discussed in the following.

We demonstrate how to apply the guideline described in the beginning of Section 6 to Theorem 2 in (Pedramfar & Aggarwal, 2024b). This allows us to obtain a generalized version of Theorems 1 in (Pedramfar & Aggarwal, 2024a). As we will discuss below, this will allow us to recover all the remaining results in Tables 1 and 2 that are marked with (*) and all the results of (Pedramfar & Aggarwal, 2024a). Note that the results of (Pedramfar & Aggarwal, 2024a) in non-stationary setting are not discussed in this paper, but they are also recovered.

We start with some definitions. Given a function class $\mathbf{F}$, we use the notation $\mathbf{F}_{\mu,\mathbf{g}}$ to denote the class of functions $q(\mathbf{y}) := \langle \mathfrak{g}(f, \mathbf{x}), \mathbf{y} - \mathbf{x} \rangle - \frac{\mu}{2} \|\mathbf{y} - \mathbf{x}\|^2 : \mathcal{K} \to \mathbb{R}$, for all $f \in \mathbf{F}$ and $\mathbf{x} \in \mathcal{K}$. This is the class of quadratic (or linear, when $\mu = 0$) functions that form the upper bound in Equation 1. Similarly, for any $B_1 > 0$, we use the notation $\mathbf{Q}_\mu[B_1]$ to denote the class of functions $q(\mathbf{y}) := \langle \mathbf{o}, \mathbf{y} - \mathbf{x} \rangle - \frac{\mu}{2} \|\mathbf{y} - \mathbf{x}\|^2 : \mathcal{K} \to \mathbb{R}$, for all $\mathbf{x} \in \mathcal{K}$ and $\mathbf{o} \in \mathbb{B}_{B_1}(\mathbf{0})$. In the following theorems, we will obtain results that allow us to reduce the problem of online optimization over $\mathbf{F}$ to the problem of online optimization over the quadratic (or linear) function class $\mathbf{F}_{\mu,\mathbf{g}}$.

**Theorem 9.** *Let $\mathcal{A}$ be algorithm for online optimization with semi-bandit feedback. Also let $\mathbf{F}$ be a differentiable function class over $\mathcal{K}$ and $\mu \geq 0$. Then the following are true.*

* *If query oracles in $\text{Adv}$ are deterministic gradient oracles, then we have*

$$\sup_{\mathcal{B} \in \text{Adv}} \mathbb{E} \left[ \max_{\mathbf{u} \in \mathcal{U}} \sum_{t=a}^{b} \left( \langle \nabla f_t(\mathbf{x}_t), \mathbf{u}_t - \mathbf{x}_t \rangle - \frac{\mu}{2} \|\mathbf{u}_t - \mathbf{x}_t\| \right) \right] \leq \mathcal{R}_{1,\text{Adv}_1^f(\mathbf{F}_{\mu,\nabla})}^{\mathcal{A}}.$$

* *On the other hand, if $\mathbf{F}$ is $M_1$-Lipschitz and query oracles in $\text{Adv}$ are stochastic gradient oracles that are bounded by $B_1 \geq M_1$, then we have*

$$\sup_{\mathcal{B} \in \text{Adv}} \max_{\mathbf{u} \in \mathcal{U}} \mathbb{E} \left[ \sum_{t=a}^{b} \left( \langle \nabla f_t(\mathbf{x}_t), \mathbf{u}_t - \mathbf{x}_t \rangle - \frac{\mu}{2} \|\mathbf{u}_t - \mathbf{x}_t\| \right) \right] \leq \mathcal{R}_{1,\text{Adv}_1^f(\mathbf{Q}_\mu[B_1])}^{\mathcal{A}}.$$

See Appendix C for proof. Note that if $f_t$ are $\mu$-strongly concave, then this result reduces to Theorem 2 in (Pedramfar & Aggarwal, 2024b). Next, we follow step (2) in the guideline to obtain the following result.

**Theorem 10.** *Let $\mathbf{F}$ be function class over $\mathcal{K}$ that is upper-quadratizable with $\mu \geq 0$, $0 < \alpha \leq 1$ and $\beta \geq 0$ and a first-order uniform wrapper $\mathcal{W}$.*

* *If $\mathcal{W}(\nabla) = \nabla$, i.e., it maps deterministic gradient oracles into deterministic gradient oracles, then we have $\mathcal{R}_{\alpha,\text{Adv}_1^f(\mathbf{F})}^{\mathcal{W}(\mathcal{A})} \leq \beta \mathcal{R}_{1,\text{Adv}_1^f(\mathbf{F}_{\mu,\nabla})}^{\mathcal{A}}.$*

* *If, for any $f \in \mathbf{F}$ and any query oracle $\mathcal{Q}_f$ bounded by $B_1$, $\mathcal{W}(\mathcal{Q}_f)$ is a stochastic query oracle for $\mathcal{W}(f)$ that is bounded by $B_1'$, then we have $\mathcal{R}_{\alpha,\text{Adv}_1^o(\mathbf{F},B_1)}^{\mathcal{W}(\mathcal{A})} \leq \beta \mathcal{R}_{1,\text{Adv}_1^f(\mathbf{Q}_\mu[B_1'])}^{\mathcal{A}}.$*

See Appendix D for proof. In this theorem, by using the uniform wrappers described in Section 7, in the special case of $i = 1$, we recover Theorems 2, 3 and 4 in (Pedramfar & Aggarwal, 2024a). (See Remarks 1 and 2) In other words, we recover all meta-algorithms in (Pedramfar & Aggarwal, 2024a) that are used to convert concave optimization algorithms into up-concave optimization algorithms.

*Remark* 6. By applying these uniform wrappers to base algorithms SO-OGA ((Garber & Kretzu, 2022)) or IA ((Zhang et al., 2018)), we recover all the results of (Pedramfar & Aggarwal, 2024a). In particular, we also recover the results for non-stationary regret described in Table 3 in (Pedramfar & Aggarwal, 2024a).

## C  PROOF OF THEOREM 9

*Proof.*

**Deterministic oracle:**

For any realization $\mathcal{B} = (\mathcal{B}_1, \cdots, \mathcal{B}_T) \in \mathrm{Adv} \subseteq \mathrm{Adv}_1^{\mathrm{f}}(\mathbf{F})$, we define $\mathcal{B}_t'(\mathbf{x}_1, \cdots, \mathbf{x}_t)$ to be the tuple $(q_t, \nabla)$ where

$$\mathcal{B}_t'(\mathbf{x}_1, \cdots, \mathbf{x}_t) := q_t := \mathbf{y} \mapsto \langle \nabla f_t(\mathbf{x}_t), \mathbf{y} - \mathbf{x}_t \rangle - \frac{\mu}{2} \|\mathbf{y} - \mathbf{x}_t\|^2,$$

and $\mathcal{B}' = (\mathcal{B}_1', \cdots, \mathcal{B}_T')$. Note that each $\mathcal{B}_t'$ is a deterministic function of $\mathbf{x}_1, \cdots, \mathbf{x}_t$ and therefore $\mathcal{B}' \in \mathrm{Adv}_1^{\mathrm{f}}(\mathbf{F}_{\mu,\nabla})$. Since the algorithm uses semi-bandit feedback, the sequence of random vectors $(\mathbf{x}_1, \cdots, \mathbf{x}_T)$ chosen by $\mathcal{A}$ is identical between the game with $\mathcal{B}$ and $\mathcal{B}'$. Hence

$$\sup_{\mathcal{B} \in \mathrm{Adv}} \mathbb{E} \left[ \max_{\mathbf{u} \in \mathcal{U}} \sum_{t=a}^{b} \left( \langle \nabla f_t(\mathbf{x}_t), \mathbf{u}_t - \mathbf{x}_t \rangle - \frac{\mu}{2} \|\mathbf{u}_t - \mathbf{x}_t\|^2 \right) \right]$$

$$= \sup_{\mathcal{B} \in \mathrm{Adv}} \mathbb{E} \left[ \max_{\mathbf{u} \in \mathcal{U}} \left( \sum_{t=a}^{b} q_t(\mathbf{u}_t) - \sum_{t=a}^{b} q_t(\mathbf{x}_t) \right) \right]$$

$$\leq \sup_{\mathcal{B}' \in \mathrm{Adv}_1^{\mathrm{f}}(\mathbf{F}_{\mu,\nabla})} \mathcal{R}_{1,\mathcal{B}'}^{\mathcal{A}} = \mathcal{R}_{1,\mathrm{Adv}_1^{\mathrm{f}}(\mathbf{F}_{\mu,\nabla})}^{\mathcal{A}}.$$

**Stochastic oracle:**

Let $\Omega^{\mathcal{Q}} = \Omega_1^{\mathcal{Q}} \times \cdots \times \Omega_T^{\mathcal{Q}}$ capture all sources of randomness in the query oracles of $\mathrm{Adv}_1^{\mathrm{o}}(\mathbf{F}, B_1)$, i.e., for any choice of $\theta \in \Omega^{\mathcal{Q}}$, the query oracle is deterministic. Hence for any $\theta \in \Omega^{\mathcal{Q}}$ and realized adversary $\mathcal{B} \in \mathrm{Adv} \subseteq \mathrm{Adv}_1^{\mathrm{f}}(\mathbf{F}, B_1)$, we may consider $\mathcal{B}_\theta$ as an object similar to an adversary with a deterministic oracle. However, note that $\mathcal{B}_\theta$ does not satisfy the unbiasedness condition of the oracle, i.e., the returned value of the oracle is not necessarily the gradient of the function at that point. Recall that $\mathcal{B}_t$ maps a tuple $(\mathbf{x}_1, \cdots, \mathbf{x}_t)$ to a tuple of $f_t$ and a stochastic query oracle for $f_t$. We will use $\mathbb{E}_{\Omega^{\mathcal{Q}}}$ to denote the expectation with respect to the randomness of query oracle and $\mathbb{E}_{\Omega_t^{\mathcal{Q}}}[\cdot] := \mathbb{E}_{\Omega^{\mathcal{Q}}}[\cdot | f_t, \mathbf{x}_t]$ to denote the expectation conditioned on the action of the agent and the adversary. Similarly, let $\mathbb{E}_{\Omega^{\mathcal{A}}}$ denote the expectation with respect to the randomness of the agent. Let $\mathbf{o}_t$ be the random variable denoting the output of $\mathcal{Q}$ at time-step $t$ and let

$$\bar{\mathbf{o}}_t := \mathbb{E}[\mathbf{o}_t \mid f_t, \mathbf{x}_t] = \mathbb{E}_{\Omega_t^{\mathcal{Q}}}[\mathbf{o}_t] = \nabla f_t(\mathbf{x}_t).$$

Similar to the deterministic case, for any realization $\mathcal{B} = (\mathcal{B}_1, \cdots, \mathcal{B}_T) \in \mathrm{Adv}$ and any $\theta \in \Omega^{\mathcal{Q}}$, we define $\mathcal{B}_{\theta,t}'(\mathbf{x}_1, \cdots, \mathbf{x}_t)$ to be the pair $(q_t, \nabla)$ where

$$q_t := \mathbf{y} \mapsto \langle \mathbf{o}_t, \mathbf{y} - \mathbf{x}_t \rangle - \frac{\mu}{2} \|\mathbf{y} - \mathbf{x}_t\|^2.$$

We also define $\mathcal{B}_\theta' := (\mathcal{B}_{\theta,1}', \cdots, \mathcal{B}_{\theta,T}')$. Note that a specific choice of $\theta$ is necessary to make sure that the function returned by $\mathcal{B}_{\theta,t}'$ is a deterministic function of $\mathbf{x}_1, \cdots, \mathbf{x}_t$ and not a random variable and therefore $\mathcal{B}_\theta'$ belongs to $\mathrm{Adv}_1^{\mathrm{f}}(\mathbf{F}_\mu[B_1])$.

Since the algorithm uses (semi-)bandit feedback, given a specific value of $\theta$, the sequence of random vectors $(\mathbf{x}_1, \cdots, \mathbf{x}_T)$ chosen by $\mathcal{A}$ is identical between the game with $\mathcal{B}_\theta$ and $\mathcal{B}_\theta'$. Therefore, for

any $\mathbf{u} \in \mathcal{U}$, we have

$$\mathbb{E}\left[\sum_{t=a}^{b}\left(\langle \nabla f_t(\mathbf{x}_t), \mathbf{u}_t - \mathbf{x}_t\rangle - \frac{\mu}{2}\|\mathbf{u}_t - \mathbf{x}_t\|^2\right)\right]$$

$$= \mathbb{E}\left[\sum_{t=a}^{b}\left(\langle \mathbb{E}\left[\mathbf{o}_t \mid f_t, \mathbf{x}_t\right], \mathbf{u}_t - \mathbf{x}_t\rangle - \frac{\mu}{2}\|\mathbf{u}_t - \mathbf{x}_t\|^2\right)\right]$$

$$= \mathbb{E}\left[\sum_{t=a}^{b}\left(\mathbb{E}\left[\langle \mathbf{o}_t, \mathbf{u}_t - \mathbf{x}_t\rangle - \frac{\mu}{2}\|\mathbf{u}_t - \mathbf{x}_t\|^2 \mid f_t, \mathbf{x}_t\right]\right)\right]$$

$$= \mathbb{E}\left[\sum_{t=a}^{b}\left(\mathbb{E}\left[q_t(\mathbf{u}_t) - q_t(\mathbf{x}_t) \mid f_t, \mathbf{x}_t\right]\right)\right]$$

$$= \mathbb{E}\left[\sum_{t=a}^{b}\left(q_t(\mathbf{u}_t) - q_t(\mathbf{x}_t)\right)\right].$$

Hence we have

$$\max_{\mathbf{u}\in\mathcal{U}}\mathbb{E}\left[\sum_{t=a}^{b}\left(\langle \nabla f_t(\mathbf{x}_t), \mathbf{u}_t - \mathbf{x}_t\rangle - \frac{\mu}{2}\|\mathbf{u}_t - \mathbf{x}_t\|\right)\right] = \max_{\mathbf{u}\in\mathcal{U}}\mathbb{E}\left[\sum_{t=a}^{b}\left(q_t(\mathbf{u}_t) - q_t(\mathbf{x}_t)\right)\right]$$

$$\leq \mathbb{E}\left[\max_{\mathbf{u}=(\mathbf{u}_1,\cdots,\mathbf{u}_T)\in\mathcal{U}}\sum_{t=a}^{b}\left(q_t(\mathbf{u}_t) - q_t(\mathbf{x}_t)\right)\right]$$

$$= \mathcal{R}_{\mathcal{B}'_\theta}^{\mathcal{A}}(\mathcal{U})[a, b]$$

where the inequality follows from Jensen's inequality. Therefore

$$\sup_{\mathcal{B}\in\text{Adv}}\max_{\mathbf{u}\in\mathcal{U}}\mathbb{E}\left[\sum_{t=a}^{b}\left(\langle \nabla f_t(\mathbf{x}_t), \mathbf{u}_t - \mathbf{x}_t\rangle - \frac{\mu}{2}\|\mathbf{u}_t - \mathbf{x}_t\|\right)\right]$$

$$\leq \sup_{\mathcal{B}\in\text{Adv}, \theta\in\Omega^{\mathcal{Q}}} \mathcal{R}_{\mathcal{B}'_\theta}^{\mathcal{A}}$$

$$\leq \sup_{\mathcal{B}'\in\text{Adv}_1^{\text{f}}(\mathbf{F}_\mu[B_1])} \mathcal{R}_{\mathcal{B}'}^{\mathcal{A}}$$

$$= \mathcal{R}_{\text{Adv}_1^{\text{f}}(\mathbf{F}_\mu[B_1])}^{\mathcal{A}} \qquad \square$$

## D    PROOF OF THEOREM 10

*Proof.*

**(i):**

We have

$$\mathcal{R}_{\alpha,\text{Adv}_1^{\text{f}}(\mathbf{F})}^{\mathcal{W}(\mathcal{A})} = \sup_{\mathcal{B}\in\text{Adv}_1^{\text{f}}(\mathbf{F})}\mathbb{E}\left[\max_{\mathbf{u}=(\mathbf{u}_1,\cdots,\mathbf{u}_T)\in\mathcal{U}}\sum_{t=a}^{b}\left(\alpha f_t(\mathbf{u}_t) - f_t(\mathcal{W}(\mathbf{x}_t))\right)\right]$$

$$\leq \sup_{\mathcal{B}\in\text{Adv}_1^{\text{f}}(\mathbf{F})}\max_{\mathbf{u}=(\mathbf{u}_1,\cdots,\mathbf{u}_T)\in\mathcal{U}}\mathbb{E}\left[\sum_{t=a}^{b}\beta\left(\langle \nabla\mathcal{W}(f_t)(\mathbf{x}_t), \mathbf{u}_t - \mathbf{x}_t\rangle - \frac{\mu}{2}\|\mathbf{u}_t - \mathbf{x}_t\|\right)\right]$$

$$= \beta\mathcal{R}_{1,\text{Adv}_1^{\text{f}}(\mathbf{H}_{\mu,\nabla})}^{\mathcal{A}}.$$

**(ii):**

Since $\mathrm{Adv}$ is oblivious, the sequence of functions $(f_1, \cdots, f_T)$ is not random and we have

$$\mathcal{R}^{\mathcal{W}(\mathcal{A})}_{\alpha, \mathrm{Adv}_1^o(\mathbf{F}, B_1)} = \sup_{\mathcal{B} \in \mathrm{Adv}_1^o(\mathbf{F}, B_1)} \mathbb{E} \left[ \max_{\mathbf{u} = (\mathbf{u}_1, \cdots, \mathbf{u}_T) \in \mathcal{U}} \sum_{t=a}^{b} \left( \alpha f_t(\mathbf{u}_t) - f_t(\mathcal{W}(\mathbf{x}_t)) \right) \right]$$

$$= \sup_{\mathcal{B} \in \mathrm{Adv}_1^o(\mathbf{F}, B_1)} \max_{\mathbf{u} = (\mathbf{u}_1, \cdots, \mathbf{u}_T) \in \mathcal{U}} \mathbb{E} \left[ \sum_{t=a}^{b} \left( \alpha f_t(\mathbf{u}_t) - f_t(\mathcal{W}(\mathbf{x}_t)) \right) \right]$$

$$\leq \sup_{\mathcal{B} \in \mathrm{Adv}_1^o(\mathbf{F}, B_1)} \max_{\mathbf{u} = (\mathbf{u}_1, \cdots, \mathbf{u}_T) \in \mathcal{U}} \mathbb{E} \left[ \sum_{t=a}^{b} \beta \left( \langle \nabla \mathcal{W}(f_t)(\mathbf{x}_t), \mathbf{u}_t - \mathbf{x}_t \rangle - \frac{\mu}{2} \| \mathbf{u}_t - \mathbf{x}_t \| \right) \right]$$

$$= \beta \mathcal{R}^{\mathcal{A}}_{1, \mathrm{Adv}_1^f(\mathbf{Q}_\mu[B_1'])}.$$

$\square$

# E  FOLLOW THE REGULARIZED LEADER

We start by defining the notion of self-concordant barrier.

**Definition 6** ((Hazan et al., 2016)). *Let $\mathcal{K} \in \mathbb{R}^d$ be a convex set with non empty interior $\mathrm{int}(\mathcal{K})$. We call a function $\Phi : \mathrm{int}(\mathcal{K}) \longrightarrow \mathbb{R}$ a $\nu$-self-concordant barrier of $\mathcal{K}$ if:*

(i) $\Phi$ *is three-times continuously differentiable, convex, and tends to infinity along any sequence of points approaching the boundary of $\mathcal{K}$;*

(ii) *For every $\mathbf{h} \in \mathbb{R}^d$ and $\mathbf{x} \in \mathrm{int}(\mathcal{K})$, we have:*

$$|\nabla^3 \Phi(\mathbf{x})[\mathbf{h}, \mathbf{h}, \mathbf{h}]| \leq 2(\nabla^2 \Phi(\mathbf{x})[\mathbf{h}, \mathbf{h}])^{3/2}, \quad |\nabla \Phi(x)[\mathbf{h}]| \leq \nu^{1/2} (\nabla^2 \Phi(\mathbf{x})[\mathbf{h}, \mathbf{h}])^{1/2}$$

*where the third-order differential is defined as $\nabla^3 \Phi(\mathbf{x})[\mathbf{h}, \mathbf{h}, \mathbf{h}] := \frac{\partial^3}{\partial t_1 \partial t_2 \partial t_3} \Phi(x + t_1 \mathbf{h} + t_2 \mathbf{h} + t_3 \mathbf{h})|_{t_1 = t_2 = t_3 = 0}$.*

Next we define the notion of local norm and dual norm with respect to a self-concordant barrier.

**Definition 7.** *For every $x \in \mathrm{int}(\mathcal{K})$, the Hessian of the self-concordant barrier induces a local norm, denoted as $\| \cdot \|_{\Phi, x}$, and a dual norm, denoted as $\| \cdot \|_{\Phi, \mathbf{x}, *}$, where for any $\mathbf{v} \in \mathbb{R}^d$,*

$$\|\mathbf{v}\|_{\Phi, \mathbf{x}} = \sqrt{\mathbf{v}^T \nabla^2 \Phi(\mathbf{x}) \mathbf{v}}, \qquad \|\mathbf{v}\|_{\Phi, \mathbf{x}, *} = \sqrt{\mathbf{v}^T (\nabla^2 \Phi(\mathbf{x}))^{-1} \mathbf{v}}.$$

An important result for FTRL is the following theorem which was proved in (Abernethy et al., 2008). It shows that if we set the regularizer to be a self-concordant barrier of $\mathcal{K}$ and the algorithm can access the unbiased estimator of $\mathbf{g}_t$, then the regret of the generated solution sequence $\{x_t\}_{t=1}^T$ can be bounded in terms of the local norm of the estimator.

**Theorem 11** ((Abernethy et al., 2008)). *Let $\mathcal{K} \subseteq \mathbb{R}^d$ be a convex set, $\Phi(x)$ be a self-concordant barrier on $\mathcal{K}$, $\{\mathbf{g}_t\}_{t=1}^T$ be a sequence of random vectors in $\mathbb{R}^d$. Then running FTRL (described in Equation 3) on a vector sequence $\{\mathbf{g}_t\}_{t=1}^T$ in $\mathbb{R}^d$ with $\Phi(x)$ as the regularizer will produce a sequence of point $\{\mathbf{x}_t\}_{t=1}^T$ in $\mathcal{K}$ where*

$$\sum_{t=1}^T \langle \mathbf{g}_t, \mathbf{y} - \mathbf{x}_t \rangle \leq \eta \sum_{t=1}^T \|\mathbf{g}_t\|_{\Phi, \mathbf{x}_t, *}^2 + \frac{\Phi(\mathbf{y}) - \Phi(\mathbf{x}_1)}{\eta},$$

*for any $\mathbf{y} \in \mathcal{K}$.*

The ellipsoid gradient estimator was proposed in (Abernethy et al., 2008), where the authors use it along with Theorem 11 to design an $\widetilde{O}(\sqrt{T})$ regret algorithm for bandit linear optimization. For a continuous function but possibly non-smooth $f : \mathbb{R}^d \to \mathbb{R}$ and an invertible matrix $\Sigma \in \mathbb{R}^{d \times d}$, we define the $\Sigma$-smoothed version of $f$.

**Definition 8.** For function $f(\mathbf{x}) : \mathbb{R}^d \to \mathbb{R}$ and invertible matrix $\Sigma \in \mathbb{R}^{d \times d}$, we call $f^\Sigma(\mathbf{x})$ a $\Sigma$-*smoothed version* of $f(\mathbf{x})$, where $f^\Sigma(\mathbf{x}) = \mathbb{E}_{\mathbf{v} \sim \mathbb{B}^d} \left[ f(\mathbf{x} + \Sigma \mathbf{v}) \right]$. Here $\mathbf{v} \sim \mathbb{B}^d$ means that $\mathbf{v}$ is sampled from the unit ball $\mathbb{B}^d$ uniformly at random.

There is a surprising fact that there is an unbiased estimator of $\nabla f^\Sigma(\mathbf{x})$ for any $\mathbf{x}$, and the estimator uses only a single query to the value oracle of $f$.

**Lemma 4** ((Abernethy et al., 2008))**.** *Let* $\Sigma \in \mathcal{R}^{d \times d}$ *be an invertible matrix,* $f(\mathbf{x}) : \mathbb{R}^d \to \mathbb{R}$ *be an arbitrary function. Then* $\nabla f^\Sigma(\mathbf{x}) = d \mathbb{E}_{\mathbf{v} \sim \mathbb{S}^{d-1}} \left[ f(\mathbf{x} + \Sigma \mathbf{v}) \Sigma^{-1} \mathbf{v} \right]$. *Here* $\mathbf{v} \sim \mathbb{S}^{d-1}$ *means that* $\mathbf{v}$ *is sampled from the* $(d-1)$-*dimensional unit sphere* $\mathbb{S}^{d-1}$ *uniformly at random.*

If $f$ is a linear function, $f^\Sigma(\mathbf{x}) = f(\mathbf{x})$, so Lemma 4 provides a one-sample unbiased estimator of the gradient of the linear function. The ellipsoid gradient estimator is usually used along with FTRL with a self-concordant regularizer $\Phi$ of $\mathcal{K}$. When the invertible matrix $\Sigma$ is set to be $(\nabla^2 \Phi(\mathbf{x}))^{-1/2}$ and $\mathbf{x} \in \text{int}(\mathcal{K})$, the sampled action $\mathbf{x} + \Sigma \mathbf{v}$ is located in the surface of a so-called **Dikin ellipsoid** centered at $\mathbf{x}$, i.e. $\{\mathbf{x}' \mid \|\mathbf{x}' - \mathbf{x}\|_{\Phi, \mathbf{x}} \leq 1\}$. The fact that Dikin ellipsoid is entirely contained in $\mathcal{K}$ allows us to define $f^\Sigma$ at $\mathbf{x}$.

We finish this section with quick overview of the concept of the Minkowski function, the Minkowski set and some of their useful properties.

**Definition 9.** Let $\mathcal{K}$ be a compact convex set, the Minkowski function $\pi_\mathbf{x} : \mathcal{K} \to \mathbb{R}$ parameterized by a pole $\mathbf{x} \in \text{int}(\mathcal{K})$ is defined as $\pi_\mathbf{x}(\mathbf{y}) := \inf\{t \geq 0 \mid x + t^{-1}(y - x) \in \mathcal{K}\}$. Given $\delta \in \mathbb{R}^+$ and $\mathbf{x}_1 \in \text{int}(\mathcal{K})$, we define the Minkowski set

$$\mathcal{K}_{\gamma, \mathbf{x}_1} := \left\{ \mathbf{x} \in \mathcal{K} \mid \pi_{\mathbf{x}_1}(\mathbf{x}) \leq (1 + \gamma)^{-1} \right\}.$$

**Lemma 5** ((Abernethy et al., 2008))**.** *Let* $\mathcal{K}$ *be a compact convex set,* $\mathbf{x} \in \text{int}(\mathcal{K})$ *with diameter* $D$, $\mathbf{u}_* \in \mathcal{K}$ *and* $\hat{\mathbf{u}}_* := \text{argmin}_{\mathbf{z} \in \mathcal{K}_{\gamma, \mathbf{x}}} \|\mathbf{z} - \mathbf{u}_*\|$ *be the projection of* $\mathbf{u}_*$ *onto the Minkowski set* $\mathcal{K}_{\gamma, \mathbf{x}}$, *then*

$$\|\mathbf{u}_* - \hat{\mathbf{u}}_*\| \leq \gamma D.$$

The following lemma provides an upper bound of the difference between the function value of a self-concordant barrier at two different points.

**Lemma 6** ((Nesterov & Nemirovskii, 1994))**.** *Let* $\Phi$ *be a* $\nu$-*self-concordant barrier over a compact convex set* $\mathcal{K}$, *then for all* $\mathbf{x}, \mathbf{y} \in \text{int}(\mathcal{K})$:

$$\Phi(\mathbf{y}) - \Phi(\mathbf{x}) \leq \nu \log \frac{1}{1 - \pi_\mathbf{x}(\mathbf{y})}.$$

# F Technical Lemmas

This section provides some technical lemmas that will be used in the proofs later.

**Lemma 7.** *Let* $\mathcal{K}$ *be a compact set and let* $f : \mathcal{K} \to \mathbb{R}^d$ *be an* $M_2$-*smooth function. Then* $f$ *may be extended to an* $M_2$-*smooth function* $\tilde{f} : \mathbb{R}^d \to \mathbb{R}$.

*Proof.* The function $\nabla F$ is an $M_2$-Lipschitz function defined on $\mathcal{K}$. Therefore, according to Kirszbraun theorem (Kirszbraun, 1934) it may be extended to a function $g : \mathbb{R}^d \to \mathbb{R}^d$ that is $M_2$-Lipschitz. Now the result follows directly from Whitney's extension theorem (Whitney, 1934). □

Parts (i)-(iii) of the following lemma are well-known in the literature. (See Lemma A.5 in (Wan et al., 2023) for a proof). Here we provide a proof for part (iv).

**Lemma 8.** *Following properties hold for* $\Sigma$-*smoothed version of a function* $f(\mathbf{x})$ *for an invertible matrix* $\Sigma$.

    *(i) If* $f(\mathbf{x})$ *is a monotone function, then so is* $f^\Sigma(\mathbf{x})$.

    *(ii) If* $f(\mathbf{x})$ *is* $M_1$-*Lipschitz, then so is* $f^\Sigma(\mathbf{x})$.

    *(iii) If* $f(\mathbf{x})$ *is* $M_2$-*smooth, then so is* $f^\Sigma(\mathbf{x})$.

*(iv) If $f$ is upper-quadratizable with a uniform wrapper $\mathcal{W}$ and $\alpha, \beta$ and $\mu$, then we have*

$$\alpha f^\Sigma(\mathbf{y}) - (f \circ \mathcal{W})^\Sigma(\mathbf{x}) \leq \beta \left( \left\langle \nabla(\mathcal{W}(f))^\Sigma(\mathbf{x}), \mathbf{y} - \mathbf{x} \right\rangle - \frac{\mu}{2} \|\mathbf{y} - \mathbf{x}\|^2 \right).$$

*Proof.* We have

$$\begin{aligned}
\alpha f^\Sigma(\mathbf{y}) - (f \circ \mathcal{W})^\Sigma(\mathbf{x}) &= \mathbb{E}_{\mathbf{v} \sim \mathbb{B}^d} \left[ \alpha f(\mathbf{y} + \Sigma\mathbf{v}) - f(\mathcal{W}(\mathbf{x} + \Sigma\mathbf{v})) \right] \\
&\leq \mathbb{E}_{\mathbf{v} \sim \mathbb{B}^d} \left[ \beta \left( \langle \nabla\mathcal{W}(f)(\mathbf{x} + \Sigma\mathbf{v}), \mathbf{y} - \mathbf{x} \rangle - \frac{\mu}{2} \|\mathbf{y} - \mathbf{x}\|^2 \right) \right] \\
&= \beta \left( \left\langle \mathbb{E}_{\mathbf{v} \sim \mathbb{B}^d} \left[ \nabla\mathcal{W}(f)(\mathbf{x} + \Sigma\mathbf{v}) \right], \mathbf{y} - \mathbf{x} \right\rangle - \frac{\mu}{2} \|\mathbf{y} - \mathbf{x}\|^2 \right) \\
&= \beta \left( \left\langle \nabla\mathbb{E}_{\mathbf{v} \sim \mathbb{B}^d} \left[ \mathcal{W}(f)(\mathbf{x} + \Sigma\mathbf{v}) \right], \mathbf{y} - \mathbf{x} \right\rangle - \frac{\mu}{2} \|\mathbf{y} - \mathbf{x}\|^2 \right) \\
&= \beta \left( \left\langle \nabla(\mathcal{W}(f))^\Sigma(\mathbf{x}), \mathbf{y} - \mathbf{x} \right\rangle - \frac{\mu}{2} \|\mathbf{y} - \mathbf{x}\|^2 \right). \qquad \square
\end{aligned}$$

**Lemma 9.** *If $f : \mathcal{K} \to \mathbb{R}$ is $M_1$-Lipschitz and $M_2$-smooth and $g : \mathcal{K} \to \mathcal{K}$ is $M_1'$-Lipschitz and $M_2'$-smooth, then $f \circ g$ is $M_1''$-Lipschitz and $M_2''$-smooth where $M_1'' := M_1 M_1'$ and $M_2'' := M_1 M_2' + M_2 M_1'^2$.*

*Proof.* We have

$$\|D(f \circ g)(\mathbf{x})\| = \|Df(g(\mathbf{x})) \cdot Dg(\mathbf{x})\| \leq M_1 M_1',$$

and therefore for all $\mathbf{x}, \mathbf{y} \in \mathcal{K}$, we have

$$\begin{aligned}
\|D(f \circ g)(\mathbf{x}) - D(f \circ g)(\mathbf{y})\| &= \|Df(g(\mathbf{x})) \cdot Dg(\mathbf{x}) - Df(g(\mathbf{y})) \cdot Dg(\mathbf{y})\| \\
&\leq \|Df(g(\mathbf{x})) \cdot Dg(\mathbf{x}) - Df(g(\mathbf{x})) \cdot Dg(\mathbf{y})\| \\
&\quad + \|Df(g(\mathbf{x})) \cdot Dg(\mathbf{y}) - Df(g(\mathbf{y})) \cdot Dg(\mathbf{y})\| \\
&= \|Df(g(\mathbf{x}))\| \|Dg(\mathbf{x}) - Dg(\mathbf{y})\| \\
&\quad + \|Df(g(\mathbf{x})) - Df(g(\mathbf{y}))\| \|Dg(\mathbf{y})\| \\
&\leq M_1 M_2' \|\mathbf{x} - \mathbf{y}\| + M_2 M_1' \|g(\mathbf{x}) - g(\mathbf{y})\| \\
&\leq (M_1 M_2' + M_2 M_1'^2) \|\mathbf{x} - \mathbf{y}\|. \qquad \square
\end{aligned}$$

## G  PROOF OF THEOREM 1

*Proof of Theorem 1.* We have

$$\begin{aligned}
&\sum_{t=1}^T \mathbb{E} \left[ f_t(\mathbf{u}_*) - f_t(\mathbf{x}_t) \right] \\
&= \sum_{t=1}^T \mathbb{E} \left[ f_t^{\delta\Sigma_t}(\hat{\mathbf{u}}_*) - f_t^{\delta\Sigma_t}(\mathbf{x}_t) \right] + \underbrace{\sum_{t=1}^T \mathbb{E} \left[ f_t^{\delta\Sigma_t}(\mathbf{u}_*) - f_t^{\delta\Sigma_t}(\hat{\mathbf{u}}_*) \right]}_{(A)} \\
&\quad + \underbrace{\sum_{t=1}^T \mathbb{E} \left[ f_t(\mathbf{u}_*) - f_t^{\delta\Sigma_t}(\mathbf{u}_*) \right]}_{(B)} + \underbrace{\sum_{t=1}^T \mathbb{E} \left[ f_t^{\delta\Sigma_t}(\mathbf{x}_t) - f_t(\mathbf{x}_t) \right]}_{(C)}
\end{aligned} \tag{4}$$

Note that, for the terms above to be well-defined, we need to be able to define $f_t^{\delta\Sigma_t}$ over $\mathcal{K}$ which requires computing $f_t$ over a set that is slightly larger than $\mathcal{K}$. Using Lemma 7, we assume that all functions $f_t$ are well-defined and $M_2$-smooth over $\mathbb{R}^d$.

*Bounding* $(A)$: Since $f_t(\mathbf{x})$ is $M_1$-Lipschitz continuous, $f_t^{\delta\Sigma_t}$ is also $M_1$-Lipschitz continuous by Lemma 8. Since $\|\hat{\mathbf{u}}_* - \mathbf{u}_*\| \le \gamma D$ by Lemma 5,

$$\sum_{t=1}^{T} \mathbb{E}\left[f_t^{\delta\Sigma_t}(\mathbf{u}_*) - f_t^{\delta\Sigma_t}(\hat{\mathbf{u}}_*)\right] \le \sum_{t=1}^{T} \mathbb{E}\left[|f_t^{\delta\Sigma_t}(\hat{\mathbf{u}}_*) - f_t^{\delta\Sigma_t}(\mathbf{u}_*)|\right]$$

$$\le \sum_{t=1}^{T} M_1\gamma D = M_1\gamma DT. \tag{5}$$

*Bounding* $(B)$: Since $f_t(\mathbf{x})$ is $M_2$-smooth, by Lemma 8, $f_t^{\delta\Sigma_t}$ is $M_2$-smooth. Thus,

$$f_t(\mathbf{u}_*) - f_t^{\delta\Sigma_t}(\mathbf{u}_*) = \mathbb{E}_{\mathbf{v}\sim\mathbb{B}^d}\left[f_t(\mathbf{u}_*) - f_t(\mathbf{u}_* + \delta\Sigma_t\mathbf{v})\right]$$

$$\le \mathbb{E}_{\mathbf{v}\sim\mathbb{B}^d}\left[-\langle\nabla f_t(\mathbf{u}_*), \delta\Sigma_t\mathbf{v}\rangle + \frac{M_2}{2}\|\delta\Sigma_t\mathbf{v}\|^2\right]$$

$$= \mathbb{E}_{\mathbf{v}\sim\mathbb{B}^d}\left[-\langle\nabla f_t(\mathbf{u}_*), \delta\Sigma_t\mathbf{v}\rangle\right] + \mathbb{E}_{\mathbf{v}\sim\mathbb{B}^d}\left[\frac{M_2}{2}\|\delta\Sigma_t\mathbf{v}\|^2\right]$$

$$= \mathbb{E}_{\mathbf{v}\sim\mathbb{B}^d}\left[\frac{M_2}{2}\|\delta\Sigma_t\mathbf{v}\|^2\right]$$

$$\le \frac{M_2\delta^2 D^2}{2}.$$

Note that in the last inequality, we used the fact that the Dikin ellipsoid centered at $\mathbf{x}_t$ is contained in $\mathcal{K}$ which implies that $\mathbf{x}_t + \Sigma_t\mathbf{v} \in \mathcal{K}$ and therefore $\|\Sigma_t\mathbf{v}\| \le D$. It follows that,

$$\sum_{t=1}^{T} \mathbb{E}\left[f_t(\hat{\mathbf{u}}_*) - f_t^{\delta\Sigma_t}(\hat{\mathbf{u}}_*)\right] \le \frac{M_2\delta^2 D^2 T}{2}. \tag{6}$$

*Bounding* $(C)$: Similarly,

$$f_t^{\delta\Sigma_t}(\mathbf{x}_t) - f_t(\mathbf{x}_t) = \mathbb{E}_{\mathbf{v}\sim\mathbb{B}^d}\left[f_t(\mathbf{x}_t + \delta\Sigma_t\mathbf{v}) - f_t(\mathbf{x}_t)\right]$$

$$\le \mathbb{E}_{\mathbf{v}\sim\mathbb{B}^d}\left[-\langle\nabla f_t(\mathbf{x}_t), \delta\Sigma_t\mathbf{v}\rangle + \frac{M_2}{2}\|\delta\Sigma_t\mathbf{v}\|^2\right] \le \frac{M_2\delta^2 D^2}{2}.$$

Therefore,

$$\sum_{t=1}^{T} \mathbb{E}\left[f_t^{\delta\Sigma_t}(\mathbf{x}_t) - f_t(\mathbf{x}_t)\right] \le \frac{M_2\delta^2 D^2 T}{2} \tag{7}$$

Putting 5,6,7 in 4, we see that

$$\sum_{t=1}^{T} \mathbb{E}\left[f_t(\mathbf{u}_*) - f_t(\mathbf{x}_t)\right] \le \sum_{t=1}^{T} \mathbb{E}\left[f_t^{\delta\Sigma_t}(\hat{\mathbf{u}}_*) - f_t^{\delta\Sigma_t}(\mathbf{x}_t)\right]$$

$$+ \alpha M_1\gamma DT + \frac{M_2\delta^2 D^2 T}{2} + \frac{M_2\delta^2 D^2 T}{2},$$

which completes the proof of the first claim.

To prove the second claim, we first use Lemma 4, with $\Sigma = \delta\Sigma_t$, to see that $\mathbb{E}\left[\mathbf{o}_t \mid \mathbf{x}_t\right] = \nabla f_t^{\delta\Sigma_t}(\mathbf{x}_t)$. On the other hand, since $\mathcal{Q}_t$ is bounded by $B_0$, we have

$$\|\mathbf{o}_t\|_{\mathbf{x}_t,*}^2 = \left\|\frac{d}{\delta}y_t\Sigma_t^{-1}\mathbf{v}_t\right\|_{\mathbf{x}_t,*}^2 = \frac{d^2}{\delta^2}|y_t|^2\mathbf{v}_t^T\Sigma_t^{-1}\left(\nabla^2\Phi(\mathbf{x}_t)\right)^{-1}\Sigma_t^{-1}\mathbf{v}_t \le \frac{d^2}{\delta^2}B_0^2\|\mathbf{v}_t\|^2 \le \frac{d^2 B_0^2}{\delta^2}$$

Hence, using Theorem 11 with $\mathbf{g}_t = \mathbf{o}_t$ and $\mathbf{y} = \hat{\mathbf{u}}_*$, we see that

$$
\begin{aligned}
\sum_{t=1}^{T} \mathbb{E}\left[\langle \nabla f_t^{\delta \Sigma_q}(\mathbf{x}_t), \hat{\mathbf{u}}_* - \mathbf{x}_t \rangle\right] &= \sum_{t=1}^{T} \mathbb{E}\left[\langle \mathbb{E}\left[\mathbf{o}_t \mid \mathbf{x}_t\right], \hat{\mathbf{u}}_* - \mathbf{x}_t \rangle\right] \\
&= \sum_{t=1}^{T} \mathbb{E}\left[\mathbb{E}\left[\langle \mathbf{o}_t, \hat{\mathbf{u}}_* - \mathbf{x}_t \rangle \mid \mathbf{x}_t\right]\right] \\
&= \mathbb{E}\left[\sum_{t=1}^{T}\langle \mathbf{o}_t, \hat{\mathbf{u}}_* - \mathbf{x}_t \rangle\right] \\
&\leq \mathbb{E}\left[\eta \sum_{t=1}^{T} \|\mathbf{o}_t\|_{\Phi, \mathbf{x}_t, *}^2 + \frac{\Phi(\hat{\mathbf{u}}_*) - \Phi(\mathbf{x}_1)}{\eta}\right] \\
&\leq \eta \sum_{t=1}^{T} \frac{d^2 B_0^2}{\delta^2} + \frac{\Phi(\hat{\mathbf{u}}_*) - \Phi(\mathbf{x}_1)}{\eta} \\
&\leq \frac{\eta d^2 B_0^2 T}{\delta^2} + \frac{\nu \log\left(\frac{1}{1-(1+\gamma)^{-1}}\right)}{\eta},
\end{aligned}
$$

where we used Lemma 6 in the last inequality. $\qquad\square$

## H  PROOF OF THEOREM 2

*Proof.* Let $\mathcal{B} \in \mathrm{Adv}$ be a realized adversary and let $f_1, \cdots, f_T$ be the sequence of functions selected by $\mathcal{B}$. Also let $\mathbf{u}_* \in \operatorname{argmax}_{\mathbf{u} \in \mathcal{K}} \sum_{t=1}^{T} f_t(\mathbf{u})$ and $\hat{\mathbf{u}}_* \in \operatorname{argmin}_{\mathbf{x} \in \mathcal{K}_{\gamma, \mathbf{x}_1}} \|\mathbf{u}_* - \mathbf{x}\|$ where $\gamma = T^{-1}$. We have

$$
\begin{aligned}
\mathcal{R}_{\alpha, \mathcal{B}}^{\mathcal{W}(\text{ZO-FTRL})} &= \sum_{t=1}^{T} \mathbb{E}\left[\alpha f_t(\mathbf{u}_*) - f_t(\mathcal{W}(\mathbf{x}_t))\right] \\
&= \sum_{t=1}^{T} \mathbb{E}\left[\alpha f_t^{\delta \Sigma_t}(\hat{\mathbf{u}}_*) - (f_t \circ \mathcal{W})^{\delta \Sigma_t}(\mathbf{x}_t)\right] + \alpha \underbrace{\sum_{t=1}^{T} \mathbb{E}\left[f_t^{\delta \Sigma_t}(\mathbf{u}_*) - f_t^{\delta \Sigma_t}(\hat{\mathbf{u}}_*)\right]}_{(A)} \\
&\quad + \alpha \underbrace{\sum_{t=1}^{T} \mathbb{E}\left[f_t(\mathbf{u}_*) - f_t^{\delta \Sigma_t}(\mathbf{u}_*)\right]}_{(B)} + \underbrace{\sum_{t=1}^{T} \mathbb{E}\left[(f_t \circ \mathcal{W})^{\delta \Sigma_t}(\mathbf{x}_t) - f_t(\mathcal{W}(\mathbf{x}_t))\right]}_{(C)}
\end{aligned}
$$

As in the proof of Theorem 1, we use Lemma 7 to extend all functions $f_t$ to $M_2$-smooth functions over $\mathbb{R}^d$ and we bound the terms (A) and (B) by $M_1 \gamma D T$ and $\frac{M_2 \delta^2 D^2 T}{2}$, respectively. To bound (C), we first use Lemma 9 to see that $f_t \circ \mathcal{W}$ is $M_2''$-smooth, where $M_2'' = M_1 M_2' + M_2 M_1'^2$. Hence, we see that

$$
\begin{aligned}
(f_t \circ \mathcal{W})^{\delta \Sigma_t}(\mathbf{x}_t) - f_t(\mathcal{W}(\mathbf{x}_t)) &= \mathbb{E}_{\mathbf{v} \sim \mathbb{B}^d}\left[f_t(\mathcal{W}(\mathbf{x}_t + \delta \Sigma_t \mathbf{v})) - f_t(\mathcal{W}(\mathbf{x}_t))\right] \\
&\leq \mathbb{E}_{\mathbf{v} \sim \mathbb{B}^d}\left[-\langle \nabla f_t(\mathcal{W}(\mathbf{x}_t)), \delta \Sigma_t \mathbf{v}\rangle + \frac{M_2''}{2}\|\delta \Sigma_t \mathbf{v}\|^2\right] \leq \frac{M_2'' \delta^2 D^2}{2}.
\end{aligned}
$$

Therefore,

$$
\sum_{t=1}^{T} \mathbb{E}\left[(f_t \circ \mathcal{W})^{\delta \Sigma_t}(\mathbf{x}_t) - f_t(\mathcal{W}(\mathbf{x}_t))\right] \leq \frac{M_2'' \delta^2 D^2 T}{2}
$$

Putting the bounds for (A), (B) and (C) together, we see that

$$
\begin{aligned}
\mathcal{R}_{\alpha,\mathcal{B}}^{\mathcal{W}(\text{ZO-FTRL})} &= \sum_{t=1}^{T} \mathbb{E}\left[\alpha f_t(\mathbf{u}_*) - f_t(\mathcal{W}(\mathbf{x}_t))\right] \\
&\leq \sum_{t=1}^{T} \mathbb{E}\left[\alpha f_t^{\delta\Sigma_t}(\hat{\mathbf{u}}_*) - (f_t \circ \mathcal{W})^{\delta\Sigma_t}(\mathbf{x}_t)\right] + \alpha M_1 \gamma D T + \frac{(\alpha M_2 + M_2'')\delta^2 D^2 T}{2} \\
&\leq \sum_{t=1}^{T} \mathbb{E}\left[\beta\langle\nabla(\mathcal{W}(f_t))^{\delta\Sigma_t}(\mathbf{x}_t), \hat{\mathbf{u}}_* - \mathbf{x}_t\rangle\right] + \alpha M_1 \gamma D T + \frac{(\alpha M_2 + M_2'')\delta^2 D^2 T}{2},
\end{aligned}
$$

$$\tag{8}$$

where the second inequality follows from Lemma 8.

To bound the remaining term, we use an argument similar to the one used in the proof of Theorem 1 again. Using Lemma 4 with $\Sigma = \delta\Sigma_t$ and the fact that $y_t$ is an unbiased sample of $\mathcal{W}(f_t)$ at $\mathbf{x}_t + \delta\Sigma_t \mathbf{v}_t$, we see that $\mathbb{E}\left[\mathbf{o}_t \mid \mathbf{x}_t\right] = \nabla(\mathcal{W}(f_t))^{\delta\Sigma_t}(\mathbf{x}_t)$. On the other hand, since $\mathcal{W}(\mathcal{Q}_t)$ is bounded by $B_0$, we have $|y_t| \leq B_0$, which implies that

$$
\|\mathbf{o}_t\|_{\mathbf{x}_t,*}^2 = \left\|\frac{d}{\delta} y_t \Sigma_t^{-1} \mathbf{v}_t\right\|_{\mathbf{x}_t,*}^2 = \frac{d^2}{\delta^2}|y_t|^2 \mathbf{v}_t^T \Sigma_t^{-1}\left(\nabla^2\Phi(\mathbf{x}_t)\right)^{-1}\Sigma_t^{-1}\mathbf{v}_t \leq \frac{d^2}{\delta^2}B_0^2\|\mathbf{v}_t\|^2 \leq \frac{d^2 B_0^2}{\delta^2}.
$$

Hence, using Theorem 11 with $\mathbf{g}_t = \mathbf{o}_t$ and $\mathbf{y} = \hat{\mathbf{u}}_*$, we see that

$$
\begin{aligned}
\sum_{t=1}^{T} \mathbb{E}\left[\beta\langle\nabla(\mathcal{W}(f_t))^{\delta\Sigma_t}(\mathbf{x}_t), \hat{\mathbf{u}}_* - \mathbf{x}_t\rangle\right] &= \beta\sum_{t=1}^{T}\mathbb{E}\left[\langle\mathbb{E}\left[\mathbf{o}_t \mid \mathbf{x}_t\right], \hat{\mathbf{u}}_* - \mathbf{x}_t\rangle\right] \\
&= \beta\sum_{t=1}^{T}\mathbb{E}\left[\mathbb{E}\left[\langle\mathbf{o}_t, \hat{\mathbf{u}}_* - \mathbf{x}_t\rangle \mid \mathbf{x}_t\right]\right] \\
&= \beta\mathbb{E}\left[\sum_{t=1}^{T}\langle\mathbf{o}_t, \hat{\mathbf{u}}_* - \mathbf{x}_t\rangle\right] \\
&\leq \beta\mathbb{E}\left[\eta\sum_{t=1}^{T}\|\mathbf{o}_t\|_{\Phi,\mathbf{x}_t,*}^2 + \frac{\Phi(\hat{\mathbf{u}}_*) - \Phi(\mathbf{x}_1)}{\eta}\right] \\
&\leq \beta\eta\sum_{t=1}^{T}\frac{d^2 B_0^2}{\delta^2} + \beta\frac{\Phi(\hat{\mathbf{u}}_*) - \Phi(\mathbf{x}_1)}{\eta} \\
&\leq \frac{\beta\eta d^2 B_0^2 T}{\delta^2} + \frac{\beta\nu\log(\frac{1}{1-(1+\gamma)^{-1}})}{\eta},
\end{aligned}
$$

where we used Lemma 6 in the last inequality. Plugging this into Equation 8 and using $M_2'' = M_1 M_2' + M_2 M_1'^2$ and $\gamma = T^{-1}$, we see that

$$
\begin{aligned}
\mathcal{R}_{\alpha,\mathcal{B}}^{\mathcal{W}(\text{ZO-FTRL})} &\leq \frac{\beta\eta d^2 B_0^2 T}{\delta^2} + \frac{\beta\nu\log(\frac{1}{1-(1+\gamma)^{-1}})}{\eta} \\
&\quad + \alpha M_1\gamma D T + \frac{\left(\alpha M_2 + M_1 M_2' + M_2 M_1'^2\right)\delta^2 D^2 T}{2} \\
&= O\left(\eta\delta^{-2}T + \eta^{-1}\log T + \delta^2 T\right). \qquad\qquad \square
\end{aligned}
$$

## I  PROOF OF THEOREM 6

*Proof.* Note that in all three cases, $\mathcal{W}^{\text{action}}$ is 1-Lipschitz and 0-smooth. Now the result for the first case follows immediately from the fact that $\mathcal{W}^{\text{M}} = \text{Id}$. Also note that for any zeroth order query

oracle $Q_f$ for a function $f \in \mathbf{F}^{\mathrm{M0}}$ and any $\mathbf{y} \in \mathcal{K}$

$$|\mathcal{W}^{\mathrm{M0}}(\mathcal{Q}_f)(\mathbf{y})| = |z^{-1}\mathcal{Q}_f(z * \mathbf{y})| \leq z^{-1} \cdot C\|z * \mathbf{y}\| = \|\mathbf{y}\| \leq D.$$

Thus the query oracle $\mathcal{W}(\mathcal{Q}_f)$ is bounded by $D$ and the assumptions of Theorem 2 are satisfied. The proof of boundedness of $\mathcal{W}^{\mathrm{NM}}(\mathcal{Q}_f)$ for any $f \in \mathbf{F}^{\mathrm{NM}}$ is similar. $\qquad\square$

## J  STOCHASTIC FULL-INFORMATION TO TRIVIAL QUERY - SFTT

In this section, we discuss the SFTT meta-algorithm (Algorithm 4 in (Pedramfar & Aggarwal, 2024a)) which converts algorithms that require full-information feedback into algorithms that have a trivial query oracle. In particular, it converts algorithms require zeroth-order full-information feedback into bandit algorithms.

We say a function class $\mathbf{F}$ is closed under convex combination if for any $f_1, \cdots, f_k \in \mathbf{F}$ and any $\delta_1, \cdots, \delta_k \geq 0$ with $\sum_i \delta_i = 1$, we have $\sum_i \delta_i f_i \in \mathbf{F}$.

**Theorem 12** (Theorem 7 and Remark 1 and Corollary 6 in (Pedramfar & Aggarwal, 2024a)). *Let $\mathcal{A}$ be an online optimization algorithm with full-information feedback and with $K$ queries at each time-step where $\mathcal{A}^{query}$ does not depend on the observations in the current round and $\mathcal{A}' = \mathrm{SFTT}(\mathcal{A})$. Then, for any $M_1$-Lipschitz function class $\mathbf{F}$ that is closed under convex combination and any $B_1 \geq M_1$, $0 < \alpha \leq 1$ and $1 \leq a \leq b \leq T$, let $a' = \lfloor(a-1)/L\rfloor + 1$, $b' = \lceil b/L\rceil$, $D = \mathrm{diam}(\mathcal{K})$ and let $\{T\}$ and $\{T/L\}$ denote the horizon of the adversary. If we also have $\mathcal{R}^{\mathcal{A}'}_{\alpha, \mathrm{Adv}^o_i(\mathbf{F},B)}(\mathcal{K}^T_\star)[a,b] = O(BT^\eta)$, $K = O(1)$ and $L = O(T^{\frac{1-\eta}{2-\eta}})$, then*

$$\mathcal{R}^{\mathcal{A}'}_{\alpha, \mathrm{Adv}^o_i(\mathbf{F},B)}(\mathcal{K}^T_\star)[a,b] = O\left(BT^{\frac{1}{2-\eta}}\right).$$

*More generally, the above result holds even if the query oracles are not bounded. Specifically, what we require is that the set of query oracles to be closed under convex combinations.*

---

**Algorithm 3:** Stochastic Full-information To Trivial query - $\mathrm{SFTT}(\mathcal{A})$

---

**Input :** base algorithm $\mathcal{A}$, horizon $T$, block size $L > K$.
**for** $q = 1, 2, \ldots, T/L$ **do**
    Let $\hat{\mathbf{x}}_q$ be the action chosen by $\mathcal{A}^{\mathrm{action}}$
    Let $(\hat{\mathbf{y}}^i_q)^K_{i=1}$ be the queries selected by $\mathcal{A}^{\mathrm{query}}$
    Let $(t_{q,1}, \ldots, t_{q,L})$ be a random permutation of $\{(q-1)L+1, \ldots, qL\}$
    **for** $t = (q-1)L+1, \ldots, qL$ **do**
        **if** $t = t_{q,i}$ *for some* $1 \leq i \leq K$ **then**
            Play the action $\mathbf{x}_t = \hat{\mathbf{y}}^i_q$
            Return the observation to the query oracle as the response to the $i$-th query
        **else**
            Play the action $\mathbf{x}_t = \hat{\mathbf{x}}_q$
        **end**
    **end**
**end**

---

## K  PROOF OF THEOREM 7

*Proof.* All three class of functions considered are closed under convex combination. Therefore we may directly apply Theorems 6 and 12 to obtain this result for the first case.

For any sequence of functions $f_1, \cdots, f_k$ and query oracles $\mathcal{Q}_1, \cdots, \mathcal{Q}_k$ for these functions that are contained within a cone $\mathrm{Cone}(\mathbf{0}, C)$ and non-negative numbers $\delta_1, \cdots, \delta_k$ such that $\sum_i \delta_i = 0$, the query oracle $\overline{Q}$ that uses $\mathcal{Q}_i$ with probability $\delta_i$ is trivially a query oracle for $\sum_i \delta_i f_i$ that is also contained within this cone. Therefore, we may apply Theorem 12 to obtain this result for the second case as well. The proof of the last case is similar. $\qquad\square$

## L  PROOF OF THEOREM 8

First we state the following simple result about OTB.

**Theorem 13** (Theorem 8 in (Pedramfar & Aggarwal, 2024a)). *If $\mathcal{A}$ is an online algorithm that queries no more than $K = T^\theta$ times per time-step and obtains an $\alpha$-regret bound of $O(T^\delta)$, then the sample complexity of $\mathrm{OTB}(\mathcal{A})$ is $\Omega(\epsilon^{-\frac{1+\theta}{1-\delta}})$.*

*Proof of Theorem 8.* This is an immediate corollary of Theorem 6 and the guarantees for the OTB meta-algorithm stated in Theorem 13. $\square$