# OpenReview forum: "Uniform Wrappers: Bridging Concave to Quadratizable Functions in Online Optimization"
_ICLR.cc/2025/Conference — Submitted to ICLR 2025_

### Official Review · Reviewer_e3zD · 2024-10-29

**Soundness:** 3
**Presentation:** 3
**Contribution:** 2
**Rating:** 3
**Confidence:** 4

**Summary:**

This article develops a general framework for transferring the algorithms and their regret guarantees from online convex/concave optimization to online quadratizable optimization. Within this framework, the authors employ a variant of Follow-the-Regularized-Leader (FTRL) from [1] to enhance the regret guarantees for several classes of weakly DR-submodular functions under zeroth-order and bandit feedback.

[1] Ankan Saha and Ambuj Tewari. Improved regret guarantees for online smooth convex optimization with bandit feedback. In Proceedings of the Fourteenth International Conference on Artificial Intelligence and Statistics, pp. 636–642. JMLR Workshop and Conference Proceedings, 2011.”

**Strengths:**

1.The authors have improved the regret bound for online optimization problems across three function classes under zeroth-order feedback, from $\tilde{O}(T^{3/4})$ to $\tilde{O}(T^{2/3})$.

2.The authors have enhanced the regret bound for online optimization problems across three function classes under bandit feedback, improving it from $\tilde{O}(T^{4/5})$ to $\tilde{O}(T^{3/4})$.

**Weaknesses:**

1.The technical contribution is quite limited. Although the upper quadratizable function is highly non-convex, many previous articles, especially [2], have shown that this upper quadratizable function satisfies a first-order variational inequality similar to that of a convex function (Lemma 1, 2, 3). Therefore, we can consider the upper quadratizable function as a special type of "quasi-concave" function. Consequently, it is quite natural for this paper to use a variant of FTRL [1] to improve the regret bounds in zero-order and bandit scenarios.

2.This paper lacks an in-depth discussion of the applications of the upper quadratizable function and a practical evaluation of the newly introduced algorithms. This statement may make readers feel that the authors are merely exploring a new function without clarifying its relevance or impact on the machine learning community.

3.From the full text and Definition 2, it appears that the primary purpose of introducing the upper quadratizable function is to study $\gamma$-weakly continuous DR-submodular functions. So, why not focus on "$\gamma$-weakly continuous DR-submodular functions" as the main subject of the paper? Can the authors provide examples of applications where the target function is upper quadratizable but not $\gamma$-weakly continuous DR-submodular functions?


[1] Ankan Saha and Ambuj Tewari. Improved regret guarantees for online smooth convex optimization with bandit feedback. In Proceedings of the Fourteenth International Conference on Artificial Intelligence and Statistics, pp. 636–642. JMLR Workshop and Conference Proceedings, 2011.

[2]Mohammad Pedramfar and Vaneet Aggarwal. From linear to linearizable optimization: A novel framework with applications to stationary and non-stationary DR-submodular optimization. arXiv preprint arXiv:2405.00065, 2024a.

**Questions:**

1. Could the authors provide a detailed personal perspective on how their technical contributions differ from those in references [1] and [2]?
2. Could you provide a detailed explanation of the applications of upper quadratizable functions and conduct an empirical evaluation of the newly proposed algorithms?
3. Can the authors provide examples of applications where the target function is upper quadratizable but not $\gamma$-weakly continuous DR-submodular functions?
4. Why not focus on "$\gamma$-weakly continuous DR-submodular functions" as the main subject of the paper?

---

> ### Author Response · Authors · 2024-11-20
>
> Thank you for your detailed review. First we would like to reemphasize what our contributions are and how they differ from those of (Pedramfar & Aggarwal, 2024a) by referring to the Official Comment above.
>
> *Strengths*
>
> Our core results are about meta-algorithms, about how one might go about obtaining new algorithms and proving them for linearizable optimization.
> Theorem 2 is simply one application of our framework. The strengths 1 and 2 that you mentioned are some applications of Theorem 2.
> In other words, they are applications of applications of our framework.
>
> Note that we also improve the sample complexity for offline optimization problems across three function classes under zeroth order feedback, detailed in Theorem 8, from $O(\epsilon^{-4})$ to $O(\epsilon^{-3})$.
>
> *Weaknesses*
>
> W1. Note that satisfying the first order inequality is the defining feature of linearizable/quadratizable functions, not a result to be proven.
> What Lemmas 1-3 demonstrate is that, in many settings, DR-submodular functions are upper-linearizable.
> Given the large body of work in optimization literature on FTRL type algorithms, it is natural to ask if a variant could work in this setting.
> However, to the best of our knowledge, any zeroth order FTRL type algorthm that performs better than $O(T^{3/4})$ is stochastic and therefore, as discussed above, can not be used as a base algorithm in (Pedramfar \& Aggarwal, 2024a) since it is neither first-order nor deterministic.
> Here we use ZO-FTRL as just one possible novel base algorithm.
> We decided against using too many applications as the paper is already quite dense.
> We leave other base algorithms that could be considered as a future work direction.
>
> W2. First we recall that linearizable/quadratizable functions are defined previously in (Pedramfar \& Aggarwal, 2024a).
> The simple fact that they generalize DR-submodularity in several settings is enough to justify their study.
> We refer to our response to W3 below for more discussions on their importance.
>
> Next we note that our main contribution can be thought of as a second order meta-result, a "result factory generator".
> Theorem 2 is an application of our framework when applied to the FTRL algorithm of (Saha \& Tewari, 2011) and is a "result factory".
> The specific FTRL type algorithms themselves are applications of Theorem 2 and therefore applications of applications of our framework.
> We decided against including applications of applications of applications of our framework as the number of possible things to discuss grows exponentially.
> Also note that the second level applications of our framework are 19+12 old results (not including other results in (Pedramfar \& Aggarwal, 2024a) that we didn't include in our tables) and 9 new results, as specified in Tables 1 and 2.
>
> If our focus was on one specific setting, say algorithms for bandit non-monotone continuous DR-submodular optimization, then it would have been reasonable to run experiments to evaluate the results.
> However, any application is so far down below the ladder of abstraction from our main contribution that it is not clear to what extent adding an experiment would meaningfully support our main result.
> We note that neither (Pedramfar \& Aggarwal, 2024a) nor (Wan et al., 2023), which are published in NeurIPS 2024 and ICML 2023 respectively and are special cases of our framework, have any experiments.
>
> We leave a detailed study of real world applications to a future work.
>
> W3.
> The optimization of continuous DR-submodular functions has become increasingly prominent in recent years.
> As shown in Lemmas 1-3, DR-submodular functions are in many settings linearizable.
> The results in (Pedramfar \& Aggarwal, 2024a) and our paper show that linearizability is the core property that can result in many SOTA algorithms and regret bounds for DR-submodular functions.
> Specifically, even if we restrict ourselves to DR-submodular functions, all the new 9 results of this paper are SOTA, beating prior works focusing on DR-submodular optimization.
> Basically, we consider a more general class of functions and we do not lose any performance.
> This is an important insight that could guide future research and focusing only on DR-submodular functions would obscure it.
>
> The only well-studied class of continuous DR-submodular functions that the framework of linearizable optimization does not cover yet is the class of non-monotone DR-submodular functions over downward closed convex sets.
> However, the moment a result similar to Lemmas 1-3 is proven for this class of functions, almost all the machinery would likely carry over and we obtain many new algorithms.
> Therefore, *any application of continuous DR-submodular maximization that is not limited to the class of non-monotone DR-submodular functions over downward closed convex sets is an application of our framework.*
>
> *Questions*
>
> Q1. See the Official Comment above.
>
> Q2. See response to W2 and W3 above.
>
> Q3 \& Q4. See response to W3 above.

---

> > ### Author Response · Authors · 2024-11-26
> >
> > We would appreciate it if you could let us know whether our responses have addressed your concerns in this paper and if you have any further comments.

---

> > ### Comment · Reviewer_e3zD · 2024-11-26
> >
> > Thank you for the detailed response. I have carefully reviewed your comments.
> >
> > I agree that this paper does provide some theoretical advancements in the field of online quadratizable optimization, particularly in the contexts of zeroth-order and bandit feedback settings. However, aligning with the previous two reviewers, my primary concern remains that the technical details of this paper are heavily dependent on the framework previously established via [1].  It appears to me that the contribution is relatively modest.
> >
> > Therefore, I maintain my current score.
> >
> > [1] Mohammad Pedramfar and Vaneet Aggarwal. From linear to linearizable optimization: A novel framework with applications to stationary and non-stationary DR-submodular optimization. arXiv preprint arXiv:2405.00065, 2024a.

---

> > > ### Author Response · Authors · 2024-12-04
> > >
> > > We have added a comment (https://openreview.net/forum?id=rbdlQE7HY7&noteId=YEdIYN6Gb4) for a more detailed comparison with the paper (Pedramfar & Aggarwal, 2024a). We hope that this clarifies the key concerns.

---

### Official Review · Reviewer_oRVx · 2024-11-03

**Soundness:** 3
**Presentation:** 3
**Contribution:** 1
**Rating:** 3
**Confidence:** 3

**Summary:**

This work gives a framework that reduces the problem of online optimization with quadratizable functions (Pedramar & Aggarwal, 2024a) to OCO. This results in state-of-the-art guarantees in several settings and improvements in some others.

**Strengths:**

- The framework provides improvements in the state-of-the-art regret bounds in several settings.
- The presentation is sufficiently clear.

**Weaknesses:**

Although there might be some small improvements in the analysis of Pedramar & Aggarwal (2024a), the framework appears to be nearly identical to that in Pedramar & Aggarwal (2024a). Nearly every technical result in the paper is due to Pedramar & Aggarwal (2024a), e.g. Lemma 1, Lemma 2, Lemma 3, ... I do not see any novelty in techniques or approach.

**Questions:**

What is the technical novelty with respect to Pedramar & Aggarwal (2024a)?

---

> ### Author Response · Authors · 2024-11-20
>
> First we would like to thank you for your review.
>
> Our novelty in this work is not in Lemma 1-3. Those lemmas are copied from (Pedramfar \& Aggarwal, 2024a) and we have referred to that paper for the proofs.
> Instead, our novelty is in our answer to the question highlighted in the first page:
> *When and how can we adapt algorithms from the (simpler) setup of online convex optimization into algorithms for online optimization over more general classes of functions?*
>
> We also note that our framework generalizes the results of (Pedramfar \& Aggarwal, 2024a).
> We do not improve the results in that case, we simply demonstrate how it can be recovered (Theorems 9 and 10) and show that it is indeed just a special case of our framework.
>
> We refer to the Official Comment above for more details on our contribution and how it compares to previous works.

---

> > ### Comment · Reviewer_oRVx · 2024-11-25
> >
> > Thank you for the detailed response. I went through the individual and global response, and I'm sticking with the score for now. I still see limited contribution with respect to prior work. To my understanding, this paper identifies a relatively small point for improvement that was missed by Pedramfar & Aggarwal (2024a) without significant changes to the larger approach or framework. Given how many results were given by the framework in Pedramfar & Aggarwal (2024a), it seems to me to be a fairly small contribution to improve on a few of these without any significant technical novelty to the larger approach (as far as I can tell).

---

> > > ### Author Response · Authors · 2024-11-26
> > >
> > > Thank you for your response.
> > >
> > > We have further added a comment (https://openreview.net/forum?id=rbdlQE7HY7&noteId=hkK8Y1m3ab) about the relevance of our work to the wider ML community and a comment (https://openreview.net/forum?id=rbdlQE7HY7&noteId=YEdIYN6Gb4) for a more detailed comparison with the paper (Pedramfar \& Aggarwal, 2024a).
> > >
> > > We look forward to any further questions or comments you may have.

---

### Official Review · Reviewer_Pu29 · 2024-11-05

**Soundness:** 3
**Presentation:** 2
**Contribution:** 2
**Rating:** 5
**Confidence:** 3

**Summary:**

This paper provides an in-depth exploration of advanced techniques for online optimization, particularly focusing on transitioning algorithms from concave optimization to more complex classes. This work contributes by introducing "uniform wrappers," a framework for adapting standard online convex optimization (OCO) algorithms to handle non-convex, quadratizable functions while preserving sublinear regret bounds.

**Strengths:**

- The paper considers a challenging and important problem in online optimization, namely, how to extend algorithms from the convex to the non-convex setting.
- The core idea of uniform wrappers allows traditional OCO algorithms to handle quadratizable functions by converting feedback and actions within a structured wrapper framework. This wrapper effectively bridges the gap between simpler convex optimization and more complex quadratizable function classes. Moreover, this framework is general and can be applied to a wide range of function classes.
- This paper obtains or matches the state of the art algorithm in several online optimization settings considered. For classes such as weakly DR-submodular functions under
zeroth order feedback, the framework provides superior regret guarantees, surpassing prior results in the field.

**Weaknesses:**

- The paper is quite dense and difficult to follow. The theoretical framework is complex, with multiple layers of definitions (e.g., quadratizability, up-super-gradients, and uniform wrappers) that might limit accessibility for a broader audience. While this complexity is necessary to cover a broad class of functions, it risks making the approach difficult to understand for readers unfamiliar with the field. The presentation could be improved by providing more intuition and examples to help readers understand the core ideas and contributions.
- The paper cites and builds upon prior work in DR-submodular and up-concave optimization (e.g., Pedramfar & Aggarwal, 2024), but it lacks a clear differentiation of its unique contributions. The proposed framework appears to be an incremental improvement rather than a groundbreaking advance. I suggest the authors to clarify the novelty and significance of their work compared to existing methods in the field.
- I am quite confused about the necessity of the uniform wrappers. The paper does not provide a clear motivation for why uniform wrappers are needed or how they improve upon existing methods. A more detailed discussion on the limitations of current approaches and how uniform wrappers address these limitations would be beneficial.

**Questions:**

- I wonder the significance of the upper quadratizable/linear functions. Why are these functions important, and how do they relate to real-world applications? It would be helpful to provide more context on the motivation behind this class of functions and why they are relevant in practice.
- What are the conditions under which uniform wrappers can be applied? Are there any limitations that restrict the applicability of this framework to certain function classes or settings?
- I am interested in the regret defined in Line 203. Can the results reduce to the dynamic regret and adaptive regret? If so, how do they compare to existing methods in these settings?

---

> ### Author Response · Authors · 2024-11-20
>
> Thank you for your detailed review.
> First we would like to reemphasize what our contributions are and how they differ from those of (Pedramfar \& Aggarwal, 2024a) by referring to the Official Comment above.
>
> *Weaknesses*
>
> W1. Note that linearizability/quadratizability and up-super-gradients are not novelties of our paper.
> Those concepts are defined in earlier works such as (Pedramfar \& Aggarwal, 2024a).
> Our main novelty are the notion of uniform wrappers and the guideline detailed in Section 6.
> We will include more intuition to make the concepts more approachable.
>
> W2 \& W3. See the Official Comment above.
> We will include a copy of this comparison in the final version.
>
> *Questions*
>
> Q1. We reemphasize that the notion of linearizable/quadratizable functions are defined in (Pedramfar \& Aggarwal, 2024a) and are not a novelty of our work.
>
> The optimization of continuous DR-submodular functions has become increasingly prominent in recent years.
> As shown in Lemmas 1-3, DR-submodular functions are in many settings upper-linearizable.
> The results in (Pedramfar \& Aggarwal, 2024a) and our paper show that upper-linearizability is the core property that can result in many SOTA algorithms and regret bounds for DR-submodular functions.
> Specifically, even if we restrict ourselves to DR-submodular functions, all the new 9 results of this paper are SOTA, beating prior works focusing on DR-submodular optimization.
> Basically, we consider a more general class of functions and we do not lose any performance.
> This is an important insight that could guide future research and focusing only on DR-submodular functions would obscure it.
>
> The only well-studied class of continuous DR-submodular functions that the framework of linearizable optimization does not cover yet is the class of non-monotone DR-submodular functions over downward closed convex sets.
> However, the moment a result similar to Lemmas 1-3 is proven for this class of functions, almost all the machinery would likely carry over and we obtain many new algorithms.
> Therefore, *any application of continuous DR-submodular maximization that is not limited to the class of non-monotone DR-submodular functions over downward closed convex sets is an application of our framework.*
>
> Q2. There are two parts to consider:
> 1. When can we apply uniform wrappers to obtain a new *candidate* algorithm?
> 2. When can we obtain appropriate regret bounds for the candidate algorithm?
>
> For the first part:
>
> If a function class is linearizable with a uniform wrapper, that wrapper may be applied to any online algorithm for convex/concave optimization to obtain new candidate algorithms for optimization of the function class.
> The only possible restrictions come from the wrapper itself, and therefore, the challenge here is in the construction of the wrapper.
> For example, the results of (Pedramfar \& Aggarwal, 2024a) may be thought of as 1st-order-to-1st-order uniform wrappers.
> In general there is no such restriction on orders.
> In fact, for simplicity, we have limited our definition of uniform wrappers to the case where it can only act on algorithms that take a single order of feedback.
> For example, imagine you have an algorithm $\mathcal{A}$ that requires both first and second order derivative of $f_t$ at some points.
> The uniform wrappers as defined right now can not wrap around $\mathcal{A}$.
> However, we could apply both the 1st-order-to-1st-order and the 2nd-order-to-2nd-order wrapper at the same time.
> In other words, we could think of the uniform wrappers described in Section 7 as 3 uniform wrappers, each one composed of a class of i-th-order-to-i-th-order uniform wrappers (for $i \geq 0$), and then we may apply any of those three uniform wrappers to $\mathcal{A}$.
>
> For the second part:
>
> The reason we are calling them *candidate* algorithms is that we do not necessarily have a proof.
> The guidelines in Section 6 describe one approach for how the proofs for the original base algorithm might be modified to obtain a proof for the new algorithm.
> Note that if one of the proofs for the original algorithm in the convex setting does not fit within the guideline, it does not mean that there are no proofs that do not fit within it.
> Moreover, there could be proofs of regret bound that do not come from our guideline at all.
> The guideline simply proposes one possible way to obtain a regret bound.
>
>
> Q3. It indeed can reduce to non-stationary notions of regret such as adaptive and dynamic regret.
> If the base algorithm has sublinear non-stationary regret, we expect the resulting algorithm to have similar regret bounds for the same notion of non-stationary regret.
> Table 2 in (Pedramfar \& Aggarwal, 2024a) is dedicated to results for such regrets where the base algorithms considered are known to have good adaptive or dynamic regret bounds.
> As far as we know, every single result for such cases can be thought of as application of the uniform wrapper framework.

---

> > ### Comment · Reviewer_Pu29 · 2024-11-21
> >
> > Thank you for your response. I agree with the authors that this paper presents new results through the introduction of a novel framework. However, I still have some reservations regarding the framework's novelty and overall significance. Thus, I will maintain my score. I will engage in further discussion with the other reviewers in the next stage.

---

> > > ### Author Response · Authors · 2024-11-26
> > >
> > > Thank you for your response.
> > >
> > > We have further added a comment (https://openreview.net/forum?id=rbdlQE7HY7&noteId=hkK8Y1m3ab) about the relevance of our work to the wider ML community and a comment (https://openreview.net/forum?id=rbdlQE7HY7&noteId=YEdIYN6Gb4) for a more detailed comparison with the paper (Pedramfar \& Aggarwal, 2024a).
> > >
> > > We look forward to any further questions or comments you may have.

---

### Author Response · Authors · 2024-11-20

Here we respond to some common comments and clarify some aspects of our contributions.

**Comparison with previous works**

The core question this paper addresses is
*When and how can we adapt algorithms from the (simpler) setup of online convex optimization into algorithms for online optimization over linearizable functions?*

Consider the following classes of online algorithms for convex/concave optimization:
I. Deterministic algorithms with deterministic first-order semi-bandit feedback.
II. All algorithms that *have* a proof which can be adapted using steps 0-2 detailed in Section 6.
III. All algorithms.
In (Pedramfar \& Aggarwal, 2024a) the notion of linearizable function classes are defined and a meta-algorithm is proposed for the class I above.
This meta-algorithm, namely OMBQ, can only be applied to class I and Theorem 1 in that paper shows that the guarantees for the original algorithm carry over to the linearizable case.
However, this class is quite limited.
For example, the requirement of first-order semi-bandit feedback rules out all zeroth order or bandit algorithms.
Or the requirement that the algorithm should be deterministic rules out algorithms such as Follow-The-Perturbed-Leader.

On the other hand, in this paper we propose meta-algorithms that could be applied to *any* algorithm.
Uniform wrappers can wrap any algorithm and propose new algorithms for linearizable optimization.
The regret guarantees do not necessarily carry over for any algorithm imaginable.
However, the guideline described in Section 6 shows how to obtain regret guarantees for algorithms belonging to class II.

To summarize, while the meta-algorithm OMBQ proposed in (Pedramfar \& Aggarwal, 2024a) applies to class I above, we propose uniform wrappers which can wrap *any* algorithm.
When it comes to regret guarantees, while Theorem 1 in (Pedramfar \& Aggarwal, 2024a) again applies to class I above, our guideline applies to class II which is significantly larger.

We achieved this by going through and gaining a deeper understanding of the existing algorithms and proofs and identifying exactly what makes them work, abstracting it away and condensing it into meta-algorithms (i.e., uniform wrappers) and the guideline detailed in Section 6.
In particular, the OMBQ meta-algorithm may be seen as an application of our guideline to Theorem 2 in (Pedramfar \& Aggarwal, 2024b) using a special case, namely 1st-order-to-1st-order case, of the uniform wrappers described in Section 7.
However, this is only a special case, while our framework is much more general.


Note that all results in Tables 1 and 2 that are marked with a blue (*) are results that fit into our framework and could be seen as an application of uniform wrappers. (The same could be said of every result of (Pedramfar \& Aggarwal, 2024a), even those that we didn't mention in Table 1.)
Moreover, the proofs of all those results also fit within our framework and can be seen as application of the guideline specified in Section 6.
Furthermore, to demonstrate the generality of our framework, besides recovering previous results, we also obtain 9 new results, all of which are SOTA within the respective setting.

In other words, we are not just proving a single result. We are demonstrating how many results in the literature (most of which are SOTA in their respective settings) can be obtained via a single framework and when and how one might go about obtaining more results.


**Specific new results**

As specific applications of our framework, we obtain 9 new results.

1. We improved the regret bound for online optimization problems across three function classes under zeroth-order feedback, detailed in Theorem 6, from $O(T^{3/4})$ to $O(T^{2/3})$.

2. We enhanced the regret bound for online optimization problems across three function classes under bandit feedback, detailed in Theorem 7, improving it from $O(T^{4/5})$ to $O(T^{3/4})$.

3. We improved the sample complexity for offline optimization problems across three function classes under zeroth order feedback, detailed in Theorem 8, improving it from $O(\epsilon^{-4})$ to $O(\epsilon^{-3})$.

There are many examples in the literature, published in top ML venues, where the authors obtain a single result in similar settings.
These applications alone are significant contributions to this field.

However, these results are merely special applications of our framework.
Specifically, Theorem 2 is an application of our framework and all the above results are applications of Theorem 2.
We will change the title of the sections to reflect this fact.

To the best of our knowledge, in all cases considered in Tables 1 and 2, i.e., both in online and offline case, whether projection-free or not, (and also in all non-stationary cases that are discussed in (Pedramfar \& Aggarwal, 2024a)), our framework obtains or matches the SOTA in DR-submodular optimization.

---

> ### Author Response · Authors · 2024-11-26
>
> **More detailed comparison with (Pedramfar \& Aggarwal, 2024a)**
>
> *Technical novelty*
> We note that this contribution of the result as compared to (Pedramfar \& Aggarwal, 2024a) is not trivial.
> The framework of (Pedramfar \& Aggarwal, 2024a) is simply one possible application of our framework when applied to Theorem 2 in (Pedramfar \& Aggarwal, 2024b) and as a result only applies to deterministic base algorithms with first order semi-bandit feedback.
> All extensions that are discussed are applied afterwards, using meta-algorithms that allow for conversion between different settings.
> The core framework itself is not capable of handling other base algorithms.
>
> If you want to take an algorithm $\mathcal{A}$ in convex optimization of convert it to an algorithm for DR-submodular optimization, the previous framework can only provide an answer if $\mathcal{A}$ is deterministic and first order with semi-bandit feedback.
> If this is not the case, then looking at the proofs in (Pedramfar \& Aggarwal, 2024a) could be more confusing than helpful (since the approach does not easily extend).
> Our framework identifies exactly what needs to change in the proof of regret bound for $\mathcal{A}$ in order to obtain a proof for regret bound of $\mathcal{W}(\mathcal{A})$.
>
> Generalizing beyond that framework requires understanding exactly how the proof works, what makes the proof work, and then abstracting it away.
> This is not a trivial task.
> Indeed, *it is not possible to generalize directly using a single data-point.*
> The framework of (Pedramfar \& Aggarwal, 2024a) is simply one data-point.
> We used both results from (Pedramfar \& Aggarwal, 2024a) and (Wan et al, 2023) as starting points to generalize beyond what each approach could achieve.
>
> *Contributions*
> As we mentioned before, the results of (Pedramfar \& Aggarwal, 2024a) are limited to base algorithms that are deterministic with deterministic first-order semi-bandit feedback.
> Our result applies to any base algorithm that has a proof of regret bound which can be adapted using steps 0-2 detailed in Section 6.
> This is a significant change.
> We achieve our results by introducing a novel and elegant definition of uniform wrappers together with a method for proving regret bounds for the resulting wrapped algorithms, which enabled a significantly improved formulation and facilitated various extensions.
> This mathematical elegance paves the way for more precise algorithmic innovations in the future, extending beyond those constrained by Theorem 2 in (Pedramfar \& Aggarwal, 2024a).
>
> There are many algorithms in the literature that simply do not fit within the framework of (Pedramfar \& Aggarwal, 2024a).
> In the following, we go over three categories of algorithms.
>
> 1. Zeroth order algorithms:
> The previous framework rules out all bandit (or even zeroth order) algorithms as base algorithms. It can only generate bandit results by applying STB meta-algorithm to another algorithm.
> Compare this to the convex case.
> In bandit convex optimization, the result of (Online convex optimization in the bandit setting: gradient descent without a gradient, Flaxman et al, 2004) could be seen as an application of STB to online gradient descent.
> All the progress made in bandit convex optimization in the last 20 years has been with algorithms that are *NOT* applications of such meta-algorithm to another algorithm.
>
> 2. Second order algorithms:
> However, all we mentioned about the bandit case are just some aspect of our contribution.
> Another aspect is any second order method.
> The previous results were limited to first order.
> They can not be applied to any second order method for convex optimization, such as Newton methods.
>
> 3. Non-deterministic algorithms:
> Furthermore, our result is still more general than what we discussed above.
> Follow-The-Perturbed-Leader is a classical algorithm in online optimization. However, it is not deterministic, it requires sampling from a noise distribution.
> There are many results in the literature about Follow-The-Perturbed-Leader type algorithms and their importance.
> However, these algorithms are just one possible example of algorithms that are not deterministic.
> Another example is the algorithm of (Efficient Projection-Free Online Convex Optimization with Membership Oracle, Mhammedi, CoLT 2022), which an efficient result that uses membership oracle (as opposed to projection oracle in online gradient descent or linear optimization oracle in Frank-Wofle type algorithms).
> Just like Follow-The-Perturbed-Leader type algorithms, this algorithm is not deterministic and therefore the framework of (Pedramfar \& Aggarwal, 2024a) simply can not be applied.
>
> Lifting the limit from *only* deterministic first-order semi-bandit algorithms as the base algorithm is not in any way a small contribution.
> The progress in the field of convex optimization over the last 20 years demonstrates the limitation of such algorithms.

---

> > ### Author Response · Authors · 2024-11-26
> >
> > **More detailed comments about the relevance of our work**
> >
> > 1. The topic of ``optimization" is explicitly stated as one of the relevant topics in ICLR.
> > 2. Other than convex setting, DR-submodular optimization is one of few classes of problems where global guarantees have been achieved in the literature.
> > 3. To see the relevance to the wider ML community, we went through some of the papers we have cited that were published in NeurIPS, ICML, ICLR, AISTATS or JMLR.
> > We found over 20 papers and the median number of algorithms that were either introduced or their regret bound obtained/improved is 3.
> > In particular, we only found 4 papers in total that had studied more than 9 algorithms (as in our paper).

---

### Public Comment · ~Mohammad_Pedramfar1 · 2026-01-16

Please see the revised version of the paper, published in NeurIPS, at https://openreview.net/forum?id=dgQRSJdo6a

---

### Meta-Review · Area_Chair_no1g · 2024-12-18

**Metareview:**

The paper introduces the notion of "uniform wrappers", an algorithmic meta-framework intended to provide regret guarantees for various classes of functions under zeroth-order feedback. The class of functions considered are essentially "quadratizable" in the sense of Pedramfar and Aggarwal (2024), which implies that its optimum points satisfy a first-order variational inequality similar to (pseudo-) convexity. The authors provide a meta-analysis based on these wrappers, which allows them to extract regret guarantees from algorithms with known performance in online convex optimization.

The reviewers raised several concerns about this paper, namely that it is densely written, difficult to follow, and incremental relative to the work of Pedramfar and Aggarwal (2024). The reviewers were unanimous in their evaluation that the paper does not clear the (admittedly high) bar for ICLR and the authors' responses did not convince them otherwise, so it was decided to make a reject recommendation.

**Additional Comments On Reviewer Discussion:**

The main points raised by the reviewers were:
1. Limited contribution over Pedramfar and Aggarwal (2024).
2. Technical and conceptual clarity of the authors' write-up.
3. Limited practical relevance of the proposed framework.

The authors' rebuttal did not meaningfully swing the reviewers' opinions on the above, hence the reject recommendation.

---

### Decision · Program_Chairs · 2025-01-22

Reject